# Review on Efficiency and Anomalies in Stock Markets

**Kai-Yin Woo [1] , Chulin Mai [2], Michael McAleer [3,4,5,6,7] and Wing-Keung Wong [8,9,10,*]**

[1] Department of Economics and Finance, Hong Kong Shue Yan University, Hong Kong 999077, China; kywoo@hksyu.edu

[2] Department of International Finance, Guangzhou College of Commerce, Guangzhou 511363, China; maichulin@foxmail.com

[3] Department of Finance, Asia University, Taichung 41354, Taiwan; michael.mcaleer@gmail.com

[4] Discipline of Business Analytics, University of Sydney Business School, Sydney, NSW 2006, Australia

[5] Econometric Institute, Erasmus School of Economics, Erasmus University Rotterdam, 3062 Rotterdam, The Netherlands

[6] Department of Economic Analysis and ICAE, Complutense University of Madrid, 28040 Madrid, Spain

[7] Institute of Advanced Sciences, Yokohama National University, Yokohama 240-8501, Japan

[8] Department of Finance, Fintech Center, and Big Data Research Center, Asia University, Taichung 41354, Taiwan

[9] Department of Medical Research, China Medical University Hospital, Taichung 40447, Taiwan

[10] Department of Economics and Finance, Hang Seng University of Hong Kong, Hong Kong 999077, China

* Correspondence: wong@asia.edu.tw

**Abstract:** The efficient-market hypothesis (EMH) is one of the most important economic and financial hypotheses that have been tested over the past century. Due to many abnormal phenomena and conflicting evidence, otherwise known as anomalies against EMH, some academics have questioned whether EMH is valid, and pointed out that the financial literature has substantial evidence of anomalies, so that many theories have been developed to explain some anomalies. To address the issue, this paper reviews the theory and literature on market efficiency and market anomalies. We give a brief review on market efficiency and clearly define the concept of market efficiency and the EMH. We discuss some efforts that challenge the EMH. We review different market anomalies and different theories of Behavioral Finance that could be used to explain such market anomalies. This review is useful to academics for developing cutting-edge treatments of financial theory that EMH, anomalies, and Behavioral Finance underlie. The review is also beneficial to investors for making choices of investment products and strategies that suit their risk preferences and behavioral traits predicted from behavioral models. Finally, when EMH, anomalies and Behavioral Finance are used to explain the impacts of investor behavior on stock price movements, it is invaluable to policy makers, when reviewing their policies, to avoid excessive fluctuations in stock markets.

**Keywords:** market efficiency; EMH; anomalies; Behavioral Finance; Winner–Loser Effect; Momentum Effect; calendar anomalies; BM effect; the size effect; Disposition Effect; Equity Premium Puzzle; herd effect; ostrich effect; bubbles; trading rules; technical analysis; overconfidence; utility; portfolio selection; portfolio optimization; stochastic dominance; risk measures; performance measures; indifference curves; two-moment decision models; dynamic models; diversification; behavioral models; unit root; cointegration; causality; nonlinearity; covariance; copulas; robust estimation; anchoring

## 1. Introduction

The efficient-market hypothesis (EMH) is one of the most important economic and financial hypotheses that have been tested over the past century. Traditional finance theory supporting EMH is based on some important financial theories, including the arbitrage principle (Modigliani and Miller 1959, 1963; Miller and Modigliani 1961), portfolio principle (Markowitz 1952a), Capital Asset Pricing Model (Treynor 1961, 1962; Sharpe 1964; Lintner 1965; Mossin 1966), arbitrage pricing theory (Ross 1976), and option pricing theory (Black and Scholes 1973). In addition, Adam Smith (Smith 1776) commented that the rational economic man will chase the maximum personal profit. When a rational economic individual comes to stock markets, they become a rational economic investor who aims to maximize their profits in stock markets.

However, an investor's rationality requires some strict assumptions. When not every investor in the stock market looks rational enough, the assumptions could be relaxed to include some "irrational" investors who could trade randomly and independently, resulting in offsetting the effects from each other so that there is no impact on asset prices (Fama 1965a).

What if those "irrational" investors do not trade randomly and independently? In this situation, Fama (1965a) and others have commented that rational arbitrageurs will buy low and sell high to eliminate the effect on asset prices caused by "irrational" investors. Fama and French (2008) pointed out that the financial literature is full of evidence of anomalies. Another school (see, for example, Guo and Wong (2016) and the references therein for more information) believes that Behavioral Finance is not caused by "irrational" investors but is caused by the existence of many different types of investors in the market.

In this paper, we review the theory and the literature on market efficiency and market anomalies. We give a brief review on market efficiency and define clearly the concept of market efficiency and the efficient-market hypothesis (EMH). We discuss some efforts that challenge the EMH. For example, we document that investors may not carry out the dynamic optimization problems required by the tenets of classical finance theory, or follow the Vulcan-like logic of the economic individual, but use rules of thumb (heuristic) to deal with a deluge of information and adopt psychological traits to replace the rationality assumption, as suggested by Montier (2004). We then review different market anomalies, including the Winner–Loser Effect, reversal effect; Momentum Effect; calendar anomalies that include January effect, weekend effect, and reverse weekend effect; book-to-market effect; value anomaly; size effect; Disposition Effect; Equity Premium Puzzle; herd effect and ostrich effect; bubbles; and different trading rules and technical analysis.

Thereafter, we review different theories of Behavioral Finance that might be used to explain market anomalies. This review is useful to academics for developing cutting-edge treatments of financial theory that EMH, anomalies, and Behavioral Finance underlie. The review is also beneficial to investors for making choices of investment products and strategies that suit their risk preferences and behavioral traits predicted from behavioral models. Finally, when EMH, anomalies, and Behavioral Finance are used to explain the impacts of investor behavior on stock price movements, it is invaluable to policy makers in reviewing their policies to avoid excessive fluctuations in stock markets.

The plan of the remainder of the paper is as follows. In Section 2, we define the concept of market efficiency clearly, review the literature on market efficiency, and discuss several models to explain market efficiency. We discuss some market anomalies in Section 3 and evaluate Behavioral Finance in Section 4. Section 5 gives some concluding remarks.

## 2. Market Efficiency

The concept of market efficiency is used to describe a market in which relevant information is rapidly impounded into the asset prices so that investors cannot expect to earn superior profits from their investment strategies. In this section, we define the concept of market efficiency clearly, review the literature of market efficiency, and discuss several models to explain market efficiency.

## 2.1. Definition of Market Efficiency

The definition of market efficiency was first anticipated in a book written by Gibson (1889), entitled *The Stock Markets of London, Paris and New York*, in which he wrote that, when "shares become publicly known in an open market, the value which they acquire may be regarded as the judgment of the best intelligence concerning them".

In 1900, a French mathematician named Louis Bachelier published his PhD thesis, *Théorie de la Spéculation* (Theory of Speculation) (Bachelier 1900). He recognized that "past, present and even discounted future events are reflected in market price, but often show no apparent relation to price changes". Hence, the market does not predict fluctuations of asset prices. Moreover, he deduced that "The mathematical expectation of the speculator is zero", which is a statement that is in line with Samuelson (1965), who explained efficient markets in terms of a martingale. The empirical implication is that asset prices fluctuate randomly, and then their movements are unpredictable. Bachelier's contribution to the origin of market efficiency was discovered when his work was published in English by Cootner (1964) and discussed in Fama (1965a, 1970).

## 2.2. Early Development in EMH

Pearson (1905) introduced the term *random walk* to describe the path taken by a drunk, who staggers in an unpredictable and random pattern. Kendall and Hill (1953) examined weekly data on stock prices and finds that they essentially move in a random-walk pattern with near-zero autocorrelation of price changes. Working (1934) and Roberts (1959) found that the movements of stock returns look like a random walk. Osborne (1959) showed that the logarithm of common stock prices follows Brownian motion and finds evidence of the square root of time rule.

If prices follow a random walk, then it is difficult to predict the future path of asset prices. Cowles (1933, 1944) and Working (1949) documented that professional forecasters cannot successfully forecast, and professional investors cannot beat the market. Granger and Morgenstern (1963) found that short-run movements of the price series obey the random-walk hypothesis, using spectral analysis, but that long-run movements do not. There is evidence of serial correlation in stock prices (Cowles and Jones 1937) which, however, could be induced by averaging (Working 1960, and Alexander 1961). Cowles (1960) reexamined the results in Cowles and Jones (1937) and still found mixed evidence of serial correlation even after correcting an error caused by averaging.

## 2.3. Recent Developments in Market Efficiency

The 2013 Nobel laureate, Eugene Fama, provided influential contributions to theoretical and empirical investigation for the recent development of market efficiency. According to Fama (1965a), an efficient market is defined as a market in which there are many rational, profit-maximizing, actively competing traders who try to predict future asset values with current available information. In an efficient market, competition among many sophisticated traders leads to a situation where actual asset prices, at any point in time, reflect the effects of all available information, and therefore, they will be good estimates of their intrinsic values.

The intrinsic value of an asset depends upon the earnings prospects of the company under study, which is not known exactly in an uncertain world, so that its actual price is expected to be above or below its intrinsic value. If the number of the competing traders is large enough, their actions should cause the actual asset price to wander randomly about its intrinsic value through offsetting mechanisms in the markets, and then the resulting successive price changes will be independent. Independent successive price changes are then consistent with the existence of an efficient market.

A market in which the prices of securities change independently of each other is defined as a random-walk market (Fama 1965a). Fama (1965b) linked the random-walk theory to the empirical study on market efficiency. The theory of random walk requires successive prices changes to be independent and to follow some probability distribution.

When the flow of news coming into the market is random and unpredictable, current price changes will reflect only current news and will be independent of past price changes. Hence, independence of successive price changes implies that the history of an asset price cannot be used to predict its future prices and increase expected profits. It is then consistent with the existence of an efficient market. Using serial correlation tests, run tests, and Alexander's (1961) filter technique, Fama (1965b) concluded that the independence of successive price changes cannot be rejected. Then, there are no mechanical trading rules based solely on the history of price changes that would make the expected profits of the market traders higher than buy-and-hold.

The random-walk theory does not specify the shape of the probability distribution of price changes, which needs to be examined empirically. Fama (1965b) found that a Paretian distribution with characteristic exponents less than 2 fit the stock market data better than the Gaussian distribution; this finding is in line with the findings of Mandelbrot (1963). Hence, the empirical distributions have more relative frequency in their extreme tails than would be expected under a Gaussian distribution while the intrinsic values change by large amounts during a very short period of time.

*2.4. EMH*

A comprehensive review of the theory and evidence on market efficiency was first provided by Fama (1970). He defined a market in which asset prices at any time fully reflect all available information as efficient and then further introduced three kinds of tests of EMH that are concerned with different sets of relevant information. They are the weak-form tests based on the past history of prices; the semi-strong tests based on all public information, including the past history of price; and finally the strong-form tests based on all private, as well as public, information.

2.4.1. Weak-Form Tests

Weak-form tests are tests used to examine whether investors can earn abnormal profits from the past data on asset prices. If successive price changes are independent and then unpredictable, it is impossible for investors to earn more than buy-and-hold. In the literature, there is evidence of random walk and independence in the successive price changes in support of weak-form market efficiency (e.g., Alexander 1961; Fama 1965a, 1965b; Fama and Blume 1966). However, Fama (1970) documented some evidence of departures from random walk with non-zero serial correlations in successive price changes on stocks (e.g., Cootner 1964; Neiderhoffer and Osborne 1966). Nevertheless, Fama (1970) recognized that rejection of random-walk model does not imply market inefficiency. The independence assumption is too restrictive and not a necessary condition for EMH because market efficiency only requires the martingale process of asset returns (Samuelson 1965) with zero expected profits to the investors.

2.4.2. Semi-Strong-Form Tests

Semi-strong-form tests involve an event study, which is used to test the adjustment speed of asset prices in response to an event announcement released to the public. An event study averages the cumulative abnormal return (CAR) of assets of interest over time, from a specified number of pre-event time periods to a specified number of post-event periods. Fama et al. (1969) provided evidence on the reaction of share prices to stock split. The market seems to expect public information, and most price adjustments are made before the event is revealed to the market, with the rest quickly and accurately adjusted after the news is released. Fama et al. (1969) concluded that their results support the EMH. Apart from stock split, other event studies on earnings announcements (Ball and Brown 1968), announcements of discount rate changes (Waud 1970) and secondary offerings of common stocks (Scholes 1972) generally provide supportive evidence for the semi-strong form of market efficiency.

### 2.4.3. Strong-Form Tests

Strong-form tests are used to assess whether professional investors have monopolistic access to all private, as well as public, information so that they can outperform the market. Jensen (1968) evaluates the performance of mutual funds over the nineteen-year period of 1945–1964, on a risk-adjusted basis. The findings indicate that the funds cannot beat the market in favor of the strong-form market efficiency, regardless of whether loading charges, management fees, and other transaction costs are ignored.

Table 1 briefly summarizes Fama's (1970) taxonomy of EMH tests.

**Table 1.** Fama's (1970) taxonomy of efficient-market-hypothesis (EMH) tests.

| Kinds of Tests | Relevant Information Sets | Methodology | Literatures |
|---|---|---|---|
| Weak form | Past history of price | Filter tests, run tests, random-walk tests | Alexander (1961); Fama (1965a, 1965b); Fama and Blume (1966) |
| Semi-strong form | Public information including past history of price | Event study | Fama et al. (1969); Ball and Brown (1968); Waud (1970); Scholes (1972) |
| Strong form | All private and public information | Mutual fund performance | Jensen (1968) |

### 2.5. Explaining Market Efficiency by Factor Models

Under the traditional framework of rational and frictionless agents, the price of a security equals its "fundamental value". This is the present value sum of expected future cash flows, where at the time of forming the expectation, the investor handles all available information properly, and the discount rate is in line with the normal acceptable preference norm. In an efficient market, no investment strategy can obtain a risk-adjusted excess average return, or higher than the average return guaranteed by its risk, which means that there is no "free lunch" (Barberis and Thaler 2003). If EMH is applied to understand the volatility of stock market prices, there should be enough evidence to justify price changes, by showing that real investment values change over time.

Taking the American stock market as an example, from 1871 to 1986, the change of the total real investment value of the stock market was measured by three factors: the change of dividend, the change of real interest rate, and the direct measure of the inter-period marginal substitution rate (Shiller 1987). In this section, we introduce some famous models, using different factors that help examine reasons behind price changes of securities or abnormal profits. Although there are abnormal profits in the market, as long as methods are provided to predict or calculate the price that brings out the abnormal returns, the abnormal profits will return to zero and make the market efficient again, with the help of rational arbitrageur's buying low and selling high.

### 2.5.1. Fama–French Three-Factor Model

Merton (1973)'s ICAPM and Ross's APT (1976), Fama and French (1993) first documented that a Three-Factor Model could be established to explain stock returns, which is more significant than ICAPM and APT. The model considers that the excess return of a portfolio (including a single stock) can be explained by its exposure to three factors: market risk premium (*RMRF*), market value factor (*SMB*, Small market capitalization Minus Big market capitalization), book-to-market ratio factor (*HML*, High book-to-market ratio Minus Low book-to-market ratio). Based on monthly data of Pakistan financial and non-financial firms from 2002 to 2016, Ali et al. (2018) showed that the Pakistani stock market is satisfactorily explained by the Three-Factor Model, especially with the addition of SMB and HML.

### 2.5.2. Carhart Four-Factor Model

There are limitations in the Fama–French Three-Factor Model, as factors like short-term reversal, medium-term momentum, volatility, skewness, gambling, and others are not considered or included. Based on the Fama–French Three-Factor Model, Carhart (1997) developed an extended version which includes a momentum factor for asset pricing in stock markets. The extra consideration is *PR1YR*, which is the return for the one-year momentum in stock returns. By applying this Four-Factor Model, Carhart (1997) claimed that it helps to explain a significant amount of variations in time series. Furthermore, the high average returns of *SMB*, *HML*, and *PR1YR* indicate that these three factors could explain the large cross-sectional variation of the average returns of stock portfolios.

### 2.5.3. Fama–French Five-Factor Asset-Pricing Model

Although Carhart's Four-Factor Model helped develop the Fama–French Three-Factor Model, Fama and French (2015) figured out a new augmentation of the Three-Factor Model by considering profitability and investment factors, which is called the Five-Factor Asset-Pricing Model, to fix more anomaly variables that cause problems to the Three-Factor Model. In addition, Fama and French (2015) concluded that the ability of the Five-Factor Model on capturing mean stock returns performs better than the Three-Factor Model.

However, the Five-Factor Model did have its drawbacks. For those low mean returns on small stocks, just like returns of those firms invest heavily regardless of low profitability, it fails to seize, and it was tested in North America, Europe, and Asia-Pacific stock markets by Fama and French (2017). Furthermore, the performance of the model is insensitive to the way in which the factors are defined. With the increase of profitability and investment factors, the value factors of the Three-Factor Model become superfluous for describing the mean return in the samples that Fama and French (2015) test.

### 2.5.4. Factor Models in Chinese Markets

Given the development of the factor model itself, Fama–French Three-Factor Model remains the benchmark in US stock markets for many years. Yet many studies copying Fama–French three factors in China's A-share market, have not achieved very satisfactory results until Liu et al.'s (2019) Chinese version Three-Factor Model. Given the strict IPO regulation rules in China's A share market, which consists of the fact that most of the smallest listed firms are targeted as potential, and reflecting the value of, shells, in order to develop this model, Liu, Stambaugh and Yuan deleted 30% of stocks at the bottom to avoid small listed firms whose values are contaminated by shell-values. Furthermore, they use EP (earning-to-price ratio) to replace BM (book-to-market ratio) due to the former capturing the value effect more accurately in China in the sense that it is the most significant factor.

Using this Chinese version of the Three-Factor Model, most reported anomalies in China's A-share market are well explained, where profitability and volatility anomalies are included. Furthermore, Liu et al. (2019) add the turnover factor PMO (Pessimistic minus Optimistic) into the model to make it a Four-Factor Model and help explain reversal and turnover anomalies. Same on the studies of China stock markets, based on the Fama–French Five-Factor Model, Li et al. (2019) developed a Seven-Factor Model by adding trading volume and turnover rates factors.

When additional factors are included, the Chinese version of the Seven-Factor Model performs well in empirical testing of herding behavior in China's A-share market, especially during three famous stock market crash periods.

Table 2 briefly summarizes factor models that explain market efficiency.

**Table 2.** Explaining market efficiency by factor models.

| Kinds of Models | Factors in Models | Literatures |
| --- | --- | --- |
| **Fama–French Three-Factor Model** | *RMRF*, *SMB,* and *HML* | Fama and French (1993) |
| **Carhart Four-Factor Model** | *RMRF*, *SMB*, *HML,* and *PR1YR* | Carhart (1997) |
| **Fama–French Five-Factor Asset Pricing Model** | *RMRF*, *SMB*, *HML*, and profitability and investment factors | Fama and French (2015) |
| **Chinese version Three-Factor Model** | *RMRF*, *SMB,* and *EP* | Liu et al. (2019) |
| **Chinese version Four-Factor Model** | *RMRF*, *SMB*, *EP,* and *PMO* | Liu et al. (2019) |
| **Chinese version Seven-Factor Model** | *RMRF*, *SMB*, *HML*, profitability factors, investment factors, trading volume, and turnover rates factors | Li et al. (2019) |

Apart from above models, there are methods measuring or explaining liquidity in stock markets. For low-frequency data, there are spread proxies, including Roll's (1984) spread (ROLL), Hasbrouck's (2009) Gibbs estimate (HASB), LOT measure provided by Lesmond et al.'s (1999) study, and others, while there are also price impact proxies, including the AMIHUD, a measure came up by Amihud (2002), the Amivest measure, or AMIVEST set up by Cooper et al. (1985), and the PASTOR estimate put up by Pástor and Stambaugh (2003). Ahn et al. (2018) studied tick data of 1183 stocks of 21 emerging markets and proved that LOT measure and AMIHUD are the best proxy among the three, respectively.

*2.6. Explaining Market Efficiency in Subdividing Areas*

There is substantial evidence to show that markets are inefficient (see, for example, Jensen 1978; Lehmann 1990; Fama 1998; Chordia et al. 2008). Loughran and Ritter (2000) suggested that multifactor models and time-series regressions should not be used to test for market efficiency.

Market efficiency has withstood the challenge of the long-term return anomaly literature. Consistent with the assumption that anomalies in market efficiency are accidental outcomes, apparent overreactions to information are as common as underreactions, and the after-event continuation of abnormal returns before an event is as frequent as the reversal after an event. Most importantly, the obvious anomalies can be methodological, and most long-term outliers, tend to disappear as technology makes sense, as market efficiency predicts (Fama 1998). Dimson and Mussavian (1998) recorded a large number of studies showing that abnormal behaviors seem inconsistent with market efficiency at first glance.

The review and analysis of market efficiency above are not the whole stories due to continuous studies are ongoing. There are supportive academics showing that EMH is significant in subdividing areas. Jena et al. (2019) proved that both volumes Put–Call Ratio and open interest Put–Call Ratio could be sufficient predictor of market returns under certain conditions. Chang et al. (2019) pointed out that the use of financial constraints has a significant positive impact on the long-term performance of companies after issuing convertible bonds.

Based on a Structure Equation Model (SEM), Li and Zhao (2019) claimed that the comprehensive effect of housing on stock investment is positive under the background of Chinese cities. Chiang (2019) studied monthly data from 15 stock markets, along with economic policy uncertainty (EPU) as a news variable, and found that news is able to predict stock market future returns. Ehigiamusoe and Lean (2019) pointed out that financial development has a long-term positive impact on economic growth, while the real exchange rate and its volatility weaken this impact. The development of the financial sector will not bring ideal economic benefits unless it is accompanied by the decline and stability of the real exchange rate.

Moreover, in some simulation cases, possibilities are found to help explain market efficiency. For example, in the experiment of Zhang and Li (2019), AR and TAR models with gamma random errors were tested on empirical volatility data of 30 stocks, with 33% of them being very suitable, indicating that the model may be a supplement to the current Gaussian random error model with appropriate adaptability. In addition, in some cases, scholars will examine and compare several methods on market efficiency explanation, before applicable practical prediction. Guo et al. (2017b) used first-order stochastic dominance and the Omega ratio in market efficiency examination and applied the theory they put forward to test the relationship between the scale of assets and real-estate investment in Hong Kong.

It is possible that scholars will find conflicts again. When studying impacts of exchange rates on stock markets, Ferreira et al. (2019) found that the exchange rate has a significant impact on Indian stock market, while there is no significant impact on European stock market. In the research of Lee and Baek (2018), although changes in oil prices do have a significant and positive impact on renewable energy stock prices in an asymmetric manner, it is a short-term impact only. Although in some of the cases, the existing models could not be considered, or quantitative analysis and modeling are still in progress, it is reasonable to postulate that, if the abnormal returns underlying anomalies are well explained, the market will become efficient with arbitrage.

Many scholars hypothesize that stock prices that are determined in efficient markets cannot be cointegrated. Nonetheless, Dwyer and Wallace (1992) showed that market efficiency or the existence of arbitrage opportunities are not related to cointegration. However, Guo et al. (2017b) developed statistics that academics and practitioners can use to test whether the market is efficient, whether or not there are any arbitrage opportunities, and whether there is any anomaly. Stein (2009) examined whether the market is efficient by both crowding and leverage factors for sophisticated investors.

Chen et al. (2011) applied cointegration and the error-correction method to obtain an interdependence relationship between the Dollar/Euro exchange rate and economic fundamentals. They found that both price stickiness and secular growth affect the exchange rate path. Clark et al. (2011) applied stochastic dominance to get efficient portfolio from an inefficient market. By applying a stochastic dominance test, Clark et al. (2016) examined the Taiwan stock and futures markets,

Lean et al. (2015) examined the oil spot and futures markets; Qiao and Wong (2015) examined the Hong Kong residential property market; and Hoang et al. (2015b) examined the Shanghai gold market. Each of these papers concluded that the markets are efficient. However, Tsang et al. (2016) examined the Hong Kong property market by using the rental yield, and concluded that the market is not efficient and there exist arbitrage opportunities in the Hong Kong property market.

## 3. Market Anomalies

There are many market anomalies that are important areas of theoretical and practical interest in Behavioral Finance. As many market anomalies have been observed that EMH cannot explain, many academics have to think of a new theory to explain market anomalies found, and this make a very important new area in finance, Behavioral Finance, which can be used to explain many market anomalies. We discuss some market anomalies in this section and discuss Behavioral Finance in the next section.

### 3.1. Winner–Loser Effect/Reversal Effect

De Bondt and Thaler (1985, 1987) found that investors are too pessimistic about the past loser portfolio and too optimistic about the past winner portfolio, resulting in the stock price deviating from its basic value. After a period of time, when the market is automatically correct, past losers are winning positive excess returns, while past winners are having negative excess returns, which support the Winner–Loser Effect. In particular, stocks used in the experiment of De Bondt and Thaler (1985) are those top 35 and those worst 35 in the long-term (five years period), then a return reversal happens in

the next three years. Thus, a new method can be advanced to predict stock returns: using reversal strategy to buy the loser portfolio in the past three to five years and sell the winner portfolio.

This strategy enables investors to obtain excess returns in the next three to five years. Further, Jegadeesh (1990) and Lehmann (1990), respectively, proved that return reversal also happens in the short-term. The representative heuristic (Tversky and Kahneman 1974), for example, people tend to rely too heavily on small samples and rely too little on large samples, inadequately discount for the regression phenomenon, and discount inadequately for selection bias in the generation or reporting of evidence (Hirshleifer 2001), can be used to explain the Winner–Loser Effect.

Thus, due to the existence of representative heuristics, investors show excessive pessimism about the past loser portfolio and excessive optimism about the past winner portfolio, that is, investors overreact to both good news and bad news. This will lead to the undervaluation of the loser portfolio price and the overvaluation of the winner portfolio price, causing them to deviate from their basic values.

*3.2. Momentum Effect*

At the moment when more and more empirical evidences are gathered to testify Winner–Loser Effect, Jegadeesh and Titman (1993) found that stock returns are positively correlated in the period of 3–12 months, i.e., the Momentum Effect. If stock returns are examined over a period of six months, the average return of the "winner portfolio" is about 9% higher than that of the "loser portfolio". Chan et al. (1996) enlarged Jegadeesh and Titman's (1993) research samples and obtained the same results.

Research conducted by Rouwenhorst (1998) showed that the Momentum Effect also exists in other developed markets and some emerging stock markets. Moskowitz and Grinblatt (1999) studied the Momentum Effect of portfolio selected by industry, and they found that the industry portfolio has significant Momentum Effect in the US stock market, and the abnormal return is larger than that of individual portfolio.

Unlike other researchers in the literature, Asness et al. (2014) challenged the existence of momentum itself, instead of explaining it by claiming the limitation of momentum. They proved that momentum return is small in size, fragmentary, under the concern of disappearing, and only applicable in short position. In the second place, momentum itself is unstable to rely on, behind which there is no theory. Last but not least, momentum might not exist or be limited by taxes or transaction costs, and it provides various results, depending on different momentum measures in a given period of time.

*3.3. Calendar Anomalies—January Effect, Weekend Effect, and Reverse Weekend Effect*

3.3.1. January Effect

The January Effect was first discovered by Wachtel (Wachtel). In a further research, Rozeff and Kinney (1976) found that the return of NYSE's stock index in January from 1904 to 1974 was significantly higher than that of other 11 months. Gultekin and Gultekin (1983) studied the stock returns of 17 countries from l959 to l979, and found that 13 of them had higher stock returns in January than in other months. Lakonishok et al. (1998) found that, between l926 and l989, the smallest 10% of stock returns exceeded those of other stocks in January. Nippani and Arize (2008) found strong evidence of the January Effect in the study of three major US market indices: corporate bond index, industrial index, and public utility index from, 1982 to 2002. However, according to Riepe (1998), the January Effect is weakening.

The most important explanations for the January Effect are the Tax-Loss Selling Hypothesis (Gultekin and Gultekin 1983) and the Window Effect Hypothesis (Haugen and Lakonishok 1988): the Tax-Loss Selling Hypothesis holds that people will sell down stocks at the end of the year, thereby offsetting the appreciation of other stocks in that year, in order to achieve the purpose of paying less tax. After the end of the year, people buy back these stocks. This collective buying and selling leads to a year-end decline in the stock market and a January rise in the stock market the following year.

The Window Effect Hypothesis holds that institutional investors want to sell losing stocks and buy profitable stocks to decorate year-end statements. This kind of trading exerts positive price pressure on profitable stocks at the end of the year, and negative pressure on losing stocks. When the selling behavior of institutional investors stops at the end of the year, the losing stocks that were depressed in the previous year will rebound tremendously in January, leading to a larger positive trend of income generation.

Sias and Starks (1997), Poterba and Weisbenner (2001), and Chen and Singal (2004) compared and analyzed the explanatory effect of the Window Effect Hypothesis and Tax-Loss Selling Hypothesis on the January Effect, and preferred the explanation of Tax-Loss Selling Hypothesis. Starks et al. (2006), based on the above research, through the study of closed-end funds of municipal bonds, further proved that the Tax-Loss Selling Hypothesis is the real reason for the January Effect.

### 3.3.2. Weekend Effect and Reverse Weekend Effect

In distinguish or test between the Weekend Effect and Reverse Weekend Effect is easy. When one gets higher returns on Friday than on Monday, it is regarded as the Weekend Effect, and when one gets higher returns on Monday rather on Friday, it is called the Reverse Weekend Effect. Weekend effects have been identified in the foreign-exchange and money markets, as well as in stock market returns by many scholars. Based on daily data from 1990 to 2010, in the world, Europe, and other countries, Bampinas et al. (2015) investigated the weekend effect of the Securitization Real Estate Index and concluded that the average return rate on Friday is significantly higher than that on other days of the week. Chan and Woo (2012) found the evidence of reverse weekend effect when Monday exhibited the highest returns for the H-shares index in Hong Kong from 3 January 2000 to 1 August 2008.

However, Olson et al. (2015) examined the US stock market with cointegration analysis and breakpoint analysis and concluded that, after the discovery of the weekend effect in 1973, the weekend effect tends to weaken and disappears in the long run. In the United States, although the effect appears to be stronger in the 1970s than in earlier or later times, there already exist various explanations for stock market behavior on weekends. For example, the regular Weekend Effect has been attributed to payment and check-clearing settlement lags.

Kamstra et al. (2000) claimed that the importance of daylight-saving-time changes indicated in their paper makes the issue something well worth sleeping on, and a matter that is as worthy of further study as to other explanations of the weekend anomaly. When the Weekend Effect weakens and disappears, the Reverse Weekend Effect will appear. In the continuous studies of Brusa et al. (2000, 2003, 2005), the Reverse Weekend Effect was found: (1) The main stock indexes have the Reverse Weekend Effect. (2) The Weekend Effect tends to be related to small firms, while the Reverse Weekend Effect tends to be related to large firms. During the period in which Reverse Weekend Effect is observed, the Monday returns of large firms tend to follow the positive Friday returns of last Friday, but they do not follow the negative Friday returns. (3) After 1988, both the broad market index and the industry index showed positive returns on Monday. Returns were regressed, with Monday as a dummy variable, in Brusa et al.'s (2011) research, and they emphasized that the Reverse Weekend Effect is widely distributed in large companies, not just a few.

### 3.4. Book-to-Market Effect/Value Anomaly

Many studies have been undertaken on the Book-to-Market (BM) effect by scholars around the world. Fama and French (1992) studied all stocks listed in NYSE, AMEX and NASDAQ from 1963 to 1990 and found that the combination with the highest BM value (value portfolio) had a monthly average return of 1.53% over the combination with the lowest BM value (charm portfolio). Wang and Xu (2004) took A-share stocks in Shanghai and Shenzhen stock markets from June 1993 to June 2002 as samples, calculated the return data of holding one, two, and three years, and considered that the BM effect exists. The same conclusion was drawn by Lam et al.'s (2019) study covering data from July 1995 to June 2015 in Chinese stock markets.

Black (1993) and MacKinlay (1995) argued that BM effect exists only in a specific sample during a specific test period, and is the result of data mining, which is not the same as what Kothari et al. (1995) found: It is the selection bias in the formation of BM combination that causes the BM effect. However, Chan et al. (1991), Davis (1994), and Fama and French (1998) tested the stock market outside the United States or during the extended test period, and still found that the BM effect existed significantly, negating the argumentation of Black (1993) and MacKinlay (1995).

Fama and French (1992, 1993, 1996) asserted that BM represents a risk factor, i.e., financial distress risk. Companies with high BM generally have poor performance in profitability, sales, and other fundamental aspects. Their financial situation is also more fragile, making their risk higher than that of companies with low BM. What is also considered is that the high return obtained by companies with high BM is only the compensation for their own high risk, and is not the unexplained anomaly. Furthermore, for the BM effect on the international level, Fama and French (1998) confirmed that a Two-Factor Model with a relative distress risk factor added could explain it rather than an international CAPM.

De Bondt and Thaler (1987) and Lakonishok et al. (1994) agreed that the BM effect is caused by investors' overreaction to company fundamentals. On the premise of confirming the positive correlation between BM and the company's fundamentals, as investors are usually too pessimistic about companies with poor fundamentals and too optimistic about companies with good fundamentals, when the overreaction is corrected, the profits of high BM companies will be higher than those of low BM companies.

### 3.5. The Size Effect

Banz (1981) found that the stock market value decreased with the increase of company size. The phenomenon that small-cap stocks earn higher returns than those calculated by CAPM (Reinganum 1981) and large-cap stocks (Siegel 1998) clearly contradicts EMH especially in January, as size of the firm and arrival of January are regarded as public information. Lakonishok et al. (1994) found that, since the stock with high P/E ratio is riskier, if P/E ratio is presumed to be known information, then this negative relationship between P/E ratio and return rate provides a considerable prediction on the latter, bringing a serious challenge to EMH.

On the contrary, Daniel and Titman (1997) claimed that BM and size only represent the preference of investors, not the determinants of returns. Due to the poor fundamentals of high BM companies and good fundamentals of low BM companies, while investors prefer to hold value stocks with good fundamentals rather than those with poor fundamentals, the return rate of companies with high BM is higher.

### 3.6. Disposition Effect

Shefrin and Statman (1985) proposed that the Disposition Effect refers to two phenomena of the stock market: The first is that investors tend to have a strong psychology of holding loss-making stocks and are not willing to realize losses; and the second is that investors tend to avoid risks before profits, thereby willing to sell stocks in order to lock in profits. In these cases, two kinds of psychology are added to describe investors where regret and embarrassment cause the first phenomenon, and arrogance leads to the second.

The Disposition Effect is one implication of extending Kahneman and Tversky's prospect theory (1979) to investments. For example, suppose an investor purchases a stock that she believes to have an expected return high enough to justify its risk. If the stock appreciates and she continues to use the purchase price as a reference point, the stock price will then be in a more concave, risk-averse part of the investor's value function. The stock's expected return may continue to justify its risk, but if the investor lowers her expectation for the stock's return somewhat, she will be likely to sell the stock. If instead of appreciating, the stock declines, its price is in the convex, risk-seeking part of the value function.

Consequently, the investor will continue to hold the stock even if its expected return falls lower than would have been necessary for her to justify its original purchase. Thus, the investor's belief about expected return must fall further, to motivate the sale of a stock that has already declined rather than one that has appreciated. Similarly, suppose an investor holds two stocks. One is up, and the other is down. If he is facing a liquidity demand and has no new information about either stock, he is more likely to sell the stock that is up (Barber and Odean 1999).

In addition, Kahneman and Tversky (1979) argued that investors are more concerned with regret than arrogance, and therefore are more willing to take no action, which leads investors to be reluctant to lose or make a profit, while those who do not sell profitable shares worry that prices will continue to rise. In other words, if one investor is not confident enough in trading stocks, he or she tends to follow investment advisers' decision or advice to buy or sell a stock, which, at least, no matter the selected stock gains or losses, he or she is not the one to be blamed and thus reduces the feeling of regret.

Locke and Mann (2015) provided evidence that professional futures floor traders appear to be subject to Disposition Effect. These traders as a group hold losing trades longer on average than gains. Their evidence also indicates that relative aversion to loss realization is related to contemporaneous and future trader relative success. Though many factors can coordinate trading (e.g., tax-loss selling, rebalancing, changing risk preference, or superior information), Barber et al. (2005) argued their empirical results are primarily driven by three behavioral factors: the representativeness heuristic, limited attention, and the disposition. When buying, similar beliefs about performance persistence in individual stocks may lead investors to buy the same stocks—a manifestation of the representativeness heuristic.

Investors may also buy the same stocks simply because those stocks catch their attention. In contrast, when selling, the extrapolation of past performance and attention play a secondary role. Attention is less of an issue for selling, since most investors refrain from short selling and can easily give attention to the few stocks they own. If investors solely extrapolated past performance, they would sell losers. However, they do not. This is because, when selling, there is a powerful countervailing factor—the Disposition Effect—a desire to avoid the regret associated with the sale of a losing investment. Thus, investors sell winners rather than losers.

Barber et al. (2008) analyzed all trades made on the Taiwan Stock Exchange between 1995 and 1999 and provided strong evidence that, in aggregate and individually, investors have a Disposition Effect; that is, investors prefer to sell winners and hold losers. The Disposition Effect exists for both long and short position, for both men and women (to roughly the same degree), and tends to decline following periods of market appreciation.

Odean (1998a, 1999) proposed an indicator to measure the degree of Disposition Effect and used this indicator to verify the strong selling, earning, and losing tendency of US stock investors. Meanwhile, Odean also found that US stock investors sold more loss-making shares in December, making the effect less pronounced because of tax avoidance. In the research on the Disposition Effect of the Chinese stock market, Zhao and Wang (2001) concluded that Chinese investors are more inclined to sell profitable stocks and continue to hold loss-making stocks, which is more serious than foreign investors.

*3.7. Equity Premium Puzzle*

The Equity Premium Puzzle, first proposed by Mehra and Prescott (1985), refers to the fact that equity yields far exceed Treasury yields. Rubinstein (1976) and Lucas (1978) showed that the stock premium could only be explained by a very high-risk aversion coefficient. Kandel and Stambaugh (1991) argued that risk aversion is actually higher than traditionally thought. However, this leads to the risk-free interest rate puzzle of Weil (1989): In order to adapt to the low real interest rate, they observed, investors can only be assumed to give preference weights equal to or higher than their current consumption in the future. This results in low or even a negative time preference rate of investors where, in practice, negative time preferences are impossible.

In order to solve the risk-free interest rate puzzle suggested by Weil (1989), Epstein and Zin (1991) further introduced the utility function of investors' first-level risk aversion attitude, which is unrelated to the risk-aversion coefficient and the cross-time substitution elasticity of consumers. With generalized expected utility proposed, this model solves the risk-free interest rate puzzle rather than the equity premium. At the same time, more and more revised versions of the utility function occur: a utility function containing the past consumer spending habits' effect shows that equity premium is due to individuals being more sensitive to the shrinking of short-term consumption (Constantinides 1990).

The consumption utility function, affected by the average consumption level of the society, is defined to explain the risk-free interest rate puzzle to a certain extent from the demand for bonds (Abel 1990). Studies that explain the Equity Premium Puzzle under certain economic conditions, apart from the catastrophic event with low probability, as studied by Rietz (1988), will increase the stock premium. In the research of Berkman et al. (2017), the expected market risk premium was successfully explained by using a measure of global political instability as an indicator of disaster risk, a profit–price ratio, and a dividend–price ratio, respectively.

Campbell and Cochrane (1999), adding the probability of the recession which will lead to future consumption levels included in the utility function, concluded the following: The increase in the probability, on the one hand, can lead to investor risk aversion increases, so they prefer a higher risk premium; on the other hand, this will increase the investor demand to meet the motives of prevention, so that the risk-free interest rate will fall.

Cecchetti et al. (2000) proposed an irrational expectation method to explain the equity premium by comparing it with the rational expectations of Campbell and Cochrane (1999). Chen (2017) explained that the Equity Premium Puzzle is due to habits formed during the life cycle of the economy, especially during recessions; for example, households develop a habit of maintaining comfortable lifestyles, which leads to macroeconomic risks that are not reflected in asset-pricing models. Not until 2019, in a model that uses age-dependent increased risk aversion but no other illogical levels of risk aversion assumptions, did DaSilva et al. (2019) obtain results consistent with US equity premium data.

In terms of the risk of labor income, which will produce losses, Heaton and Lucas (1996, 1997) claimed that investors require higher equity premium as compensations, so that they are willing to hold stocks, then generating equity premium. However, Constantinides and Duffie (1996) argued that the corresponding situation happens at a time of economic depression, when investors are more reluctant to hold stocks, for the fear of decline in the value of their equity assets; thus, higher equity premium is necessary to attract investors.

Kogan et al. (2007) found out that equity premium could be achieved in an economy that imposes borrowing constraints, while Constantinides et al. (2002) noted that equity premium is determined by middle-aged investors under the conditions of relaxed lending constraints. Bansal and Coleman (1996) claimed that negative liquidity premium of bonds reduces the risk-free interest rate and further expands the gap with stock returns, causing the Equity Premium Puzzle.

De Long et al. (1990) claimed that dividend generation is a high-risk process that leads to a high equity premium. Lacina et al. (2018) got rid of the use of the way forecasts, proving a near-zero risk premium. In addition, individual income tax rates (McGrattan and Prescott 2010), GDP growth (Faugere and Erlach 2006), and information (Gollier and Schlee 2011; Avdisa and Wachter 2017) and spatial dominance (Lee et al. 2015) are also used to explain the Equity Premium Puzzle.

With the rise of Behavioral Finance, some scholars began to use theories from Behavioral Finance to explain the Equity Premium Puzzle. Benartzi and Thaler (1995) proposed a causal relationship between loss aversion and equity premium based on prospect theory: Precisely due to the fact that investors are afraid of stock losses, equity premium is an important factor to attract investors to hold stock assets and maintain the proportion of stocks and bonds in their portfolios.

Furthermore, Barberis et al. (2001) emphasized in the BHS model they constructed that investors' loss aversion would constantly change, thus generating equity premium, while Ang et al. (2005), Xie et al. (2016), and others explained the Equity Premium Puzzle by introducing disappointment

aversion of Behavioral Finance as an influence factor. Hamelin and Pfiffelmann (2015) used Behavioral Finance to explain why entrepreneurs who are aware of their high exposure still accept low returns and show the cognitive traders the riddle of how to explain the private equity.

Mehra and Prescott (2003) analyzed 107 papers on the research of the Equity Premium Puzzle, and drew a conclusion that none has provided a plausible explanation. Given the above review in this paper, a conclusion can be drawn that, with the existing and unsolved anomalies in stock markets, efficiency in stock markets requires certain assumptions. In other words, on the way to solve and explain anomalies, a large number of models will be set up, along with new assumptions inside those models. Many long-standing puzzles can already be solved with different techniques (Ravi 2018), though extra efforts need to be paid on academic research as the world grows quicker with technological developments, making economics complicated.

### 3.8. Herd Effect and Ostrich Effect

Patel et al. (1991) introduced two behavioral hypotheses to help explain financial phenomena: Barn Door Closing for mutual fund purchases and Herd Migration Behavior for debt–equity ratio. Barn Door Closing, in the horse protection sense, refers to undertaking behavior today that would have been profitable yesterday. Herd Migration in finance occurs when market conditions change, so that individual decision makers wish to alter their holdings substantially.

Their transition is slowed because they seek protection by traveling with the herd. Herd behavior (i.e., people will do what others are doing rather than what is optimal, given their own information) refers to behavior patterns that are correlated across individuals—but could also be caused by correlated information arrival in independently acting investors.

Herding is closely linked to impact expectations, fickle changes without new information, bubbles, fads, and frenzies. Barber et al. (2003) compared the investment decisions of groups (stock clubs) and individuals. Both individuals and clubs are more likely to purchase stocks that are associated with good reasons (e.g., a company that is featured on a list of most-admired companies). However, stock clubs favor such stocks more than individuals, despite the fact that such reasons do not improve performance. The mentioned Seven-Factor Model by Li et al. (2019) also indicates that herd behavior of Chinese A-share market is more prevalent in times of market turmoil, especially when the market falls.

Hon (2015b) found a significant correlation between the reason given by small investors for changing their current security holdings, and the reason given for the sharp correction in the bank stock market. This empirical finding suggests that herding behavior occurred frequently among the small investors, and they tend to sell their stock during the sharp correction period. Hon (2013d) found that there was a change in the behavior of small investors during and immediately after the buoyant stock market of January 2006 to October 2007, in Hong Kong. During the buoyant market, small investors were overconfident and bought stock. The small investors also exhibit herd behavior, and, once the sharp correction to the market began after October 2007, they sold the stock.

In Galai and Sade (2006)'s paper, it is recorded that government Treasury bonds provide higher maturity rates than non-current assets with the same risk level in Israel. Additional research shows that liquidity is positively correlated with market information flows. As ostriches are thought to deal with obvious risk situations by hopefully pretending that risk does not exist, so the ostrich effect is used to describe the above investors' behavior. Karlsson et al. (2009) presented a decision theoretical model in which information collection is linked to investor psychology.

For a wide range of plausible parameter values, the model predicts that the investor should collect additional information conditional on favorable news, and avoid information following bad news. Empirical evidence collected from Scandinavian investors supports the existence of the ostrich effect in financial markets.

### 3.9. Bubbles

The first study to report bubbles in experimental asset markets was published by Smith et al. (1988). Bubbles feature large and rapid price increases which result in the rising of share prices to unrealistically high levels. Bubbles typically begin with a justifiable rise in stock prices. The justification may be a technological advance or a general rise in prosperity. The rise in share prices, if substantial and prolonged, leads to members of the public believing that prices will continue to rise.

People who do not normally invest begin to buy shares in the belief that prices will continue to rise. More and more people, typically those who have no knowledge of financial markets, buy shares. This pushes up prices even further. There is euphoria and manic buying. This causes further price rises.

There is a self-fulfilling prophecy wherein the belief that prices will rise brings about the rise, since it leads to buying. People with no knowledge of investment often believe that if share prices have risen recently, those prices will continue to rise in the future (Redhead 2003). A speculative bubble can be described as a situation in which temporarily high prices are sustained largely by investors' enthusiasm rather than by consistent estimations of real value. The essence of a speculative bubble is a sort of feedback, from price increases to increased investor enthusiasm, to increased demand, and hence further price increases. According to the adaptive expectations' version of the feedback theory, feedback takes place because past price increases generate expectations of further price increases.

According to an alternative version, feedback occurs as a consequence of increased investor confidence in response to past price increases. A speculative bubble is not indefinitely sustainable. Prices cannot go up forever, and when price increases end, the increased demand that the price increases generated ends. A downward feedback may replace the upward feedback.

Shiller's (2000) paper presents evidence on two types of investor attitudes that change in important ways through time, with important consequences for speculative markets. The paper explores changes in bubble expectations and investor confidence among institutional investors in the US stock market at six-month intervals for the period 1989 to 1998 and for individual investors at the start and end of this period.

Since current owners believe that they can resell the asset at an even higher price in the future, bubbles refer to asset prices that exceed an asset's fundamental value. There are four main strands of models that identify conditions under which bubbles can exist.

The first class of models assumes that all investors have rational expectations and identical information. These models generate the testable implication that bubbles have to follow an explosive path. In the second category of models, investors are asymmetrically informed and bubbles can emerge under more general conditions because their existence need not be commonly known.

A third strand of models focuses on the interaction between rational and behavioral traders. Bubbles can persist in these models since limits to arbitrage prevent rational investors from eradicating the price impact of behavioral traders.

In the final class of models, bubbles can emerge if investors hold heterogeneous beliefs, potentially due to psychological biases, and they agree to disagree about the fundamental value. Experiments are useful to isolate, distinguish, and test the validity of different mechanisms that can lead to or rule out bubbles (Abreu and Brunnermeier 2003).

West (1987) suggested that the set of parameters needed to calculate the expected present discounted value of a stream of dividend can be estimated in two ways. One may test for speculative bubbles, or fads, by testing whether the two estimates are the same. When the test is applied to some annual US stock market data, the data usually reject the null hypothesis of no bubbles. The test is generally interesting, since it may be applied to a wide class of linear rational expectations models. The seeming tendency for self-fulfilling rumors about potential stock price fluctuations to result in actual stock price movements has long been noted by economists.

For example, in a famous passage, Keynes describes the stock market as a certain type of beauty contest: speculators devote their "intelligence to anticipating what average opinion expects average

opinion to be". In recent rational expectations' work, this possibility has been rigorously formalized, and the self-fulfilling rumors have been dubbed speculative bubbles.

Craine (1993) suggests that the fundamental value of a stock is the sum of the expected discounted dividend sequence. Bubbles are deviations in the stock's price from the fundamental value. Rational bubbles satisfy an equilibrium pricing restriction, implying that agents expect them to grow fast enough to earn the expected rate of return. The explosive growth causes the stock's price to diverge from its fundamental value.

Whether the actual volatility of equity returns is due to time variation in the rational equity risk premium or to bubbles, fads, and market inefficiencies is an open issue. Bubble tests require a well-specified model of equilibrium expected returns that have yet to be developed, and this makes inference about bubbles quite tenuous.

Chan and Woo (2008) employed a new test to detect the existence of stochastic explosive root bubbles. If a speculative bubble exists, the residual process from the regression of stock prices on dividends will not be stationary. The data series include the monthly aggregate stock price indices, dividend yields and price indices for the stock markets of Taiwan, Malaysia, Indonesia, the Philippines, Thailand, and South Korea. The sample period spans from March 1991 to October 2005 for all markets.

The dividend series are estimated by multiplying the price indices by dividend yields. The stock price indices and dividends are deflated by the producer price index for Malaysia, and by the consumer price indices for the other markets. They found evidence of bubble in stock markets of Taiwan, Malaysia, Indonesia, the Philippines and Thailand, but no evidence of bubbles in South Korea over their sample period.

Homm and Breitung (2012) proposed several reasonable bubble-testing methods, which are applied to NASDAQ index and other financial time series, to test their power properties, covering changes from random walk to explosion process. They concluded that a Chow-type break test provides the highest power and performs well relative to the power envelope, and they also put forward a program to monitor speculative bubbles in real time.

In order to explore bubbles further, in the next section, we introduce several factors underlying the bubble that help explain bubbles.

### 3.9.1. The Internet

Investors, in general, and online investors, in particular, now make decisions in a very different environment than investors in the past. They have access to far more data via the Internet. They often act without personal intermediaries. They can conduct extensive searches and comparisons on a wide variety of criteria. A critical and largely unexplored research question is how this different environment affects the decision-making of investors (Barber and Odean 2001b).

Barber and Odean (2002) analyzed 1607 investors who switched from phone-based to online trading during the 1990s. Those who switched to online trading performed well prior to going online, beating the market by more than 2% annually. After going online, they traded more actively, more speculatively, and less profitably than before, lagging the market by more than 3% annually.

Reductions in market frictions (lower trading costs, improved execution speed, and greater ease of access) do not explain these findings. Overconfidence—augmented by self-attribution bias and the illusions of knowledge and control—can explain the increase in trading and reduction in performance of online investors.

### 3.9.2. Derivatives

Hon (2013a) attempted to identify the ways that the Hong Kong companies in the Hang Seng Index Constituent Stocks manage their financial risk with derivatives. By analyzing the companies' annual reports and financial reviews, it was found that 82.6% of these companies used derivatives in 2010. Specifically, 58.7% of them used swaps to hedge interest rate risk, and 54.3% of them used

forward contracts to hedge foreign-exchange risk. The results are largely consistent with the prediction that companies use derivatives to manage their financial risk.

By investing in stocks, bonds, and other financial assets, people have been able to build up a buffer in case of being dismissed. Firms have tilted their compensation packages to management away from fixed salaries toward participation and result-based compensations, such as stock options. With such options, management has an incentive to do everything possible to boost share prices. They have an incentive to maintain an appearance of corporate success and a corporation working its way toward an impressive future with increasing profits. It seems as a strategy to boost the stock value and to refine the company's objectives and announcing that it was a part of the e-business society.

Heath et al. (1999) investigated stock-option-exercise decisions by over 50,000 employees at seven corporations. Controlling for economic factors, psychological factors influence exercise. Consistent with psychological models of beliefs, employee exercise in response to stock price trends—exercise is positively related to stock returns during the preceding month and negatively related to returns over longer horizons. Consistent with psychological models of values that include reference points, employee exercise activity roughly doubles when the stock price exceeds the maximum price attained during the previous year. Options have no purchase price to serve as a reference point.

Employees do not purchase options; they receive them at a strike price that is equal to the stock price on the date of the grant. Because employees can only exercise their options when the stock price exceeds the strike price, reference points, if they exist, will be dynamically determined by stock price movement after the grant.

CEO compensation has grown dramatically. Average CEO compensation as a multiple of average worker compensation rose from 45 in 1980, to 96 in 1990, and to an astounding 458 in 2000. A large portion of this compensation comes in the form of stock options. Economists fear that managers will behave more conservatively than is in the best interests of shareholders because managers' careers are tied to the firm. Executive stock options mitigate this problem by rewarding managers when the firm's share price goes up but not punishing them when it goes down. Such convex compensation contracts encourage managers to take risks.

Gervais et al. (2011) argued that executives are likely to be overconfident and optimistic, and thus biased, when assessing projects, and that many shareholders are under-diversified and do care about specific risk. A manager may further manipulate investor expectations by managing earnings through discretionary accounting choices. Furthermore, research indicates that earnings manipulations can affect prices.

Derivatives are a new segment of secondary market operations in India. Ganesan et al. (2004) found that a buoyant and supporting cash market is a must for a robust derivative market. Hon's (2015a) found that the majority of respondents who invested in their derivative investments during January 2013 to January 2014 in Hong Kong were relatively younger. More than 58.1% of the respondents had tertiary education for their derivatives investments. Males preferred to invest in warrants more than females did, while females preferred to invest in stock options more than males did.

Hon (2015c), based on the survey results, derived the ascending order of importance of reference group, return performance, and personal background (reference group is the least important and personal background is the most important) in the Hong Kong derivatives markets. The results of Hon's paper (Hon 2013c) indicate that small investors mostly tend to trade Callable Bull/Bear Contacts (35% of total) and warrants (23% of total). Hon (2012) identified five factors that capture the behavior of small investors in derivatives markets in Hong Kong. The factors are personal background, reference group, return performance, risk tolerance, and cognitive style.

### 3.9.3. Feedback Models

The essence of a speculative bubble is the familiar feedback pattern—from price increases to increased investor enthusiasm to increased demand and, hence, to further price increase. The higher

demand for the asset is generated by the public's memory of high past returns and optimism the high returns generate for the future (Shiller 2002).

When speculative prices go up, creating successes for some investors, this may attract public attention, promote word-of-mouth enthusiasm, and heighten expectations for further price increases. The talk attracts attention that justifies the price increases. This process, in turn, increases investor demand and thus generates another round of price increases. If the feedback is not interrupted, it may produce after many rounds a speculative "bubble", in which high expectations for further price increases support very high current prices. The high prices are ultimately not sustainable, since they are high only because of expectations of further price increases, and so the bubble eventually bursts, and prices come falling down.

The feedback that propelled that bubble carries the seeds of its own destruction, and the end of the bubble may be unrelated to news stories about fundamentals. The same feedback may also produce a negative bubble, downward price movements propelling further downward price movements, promoting word-of-mouth pessimism, until the market reaches an unsustainably low level (Shiller 2003).

### 3.9.4. Smart Money

The efficient markets theory, as it is commonly expressed, asserts that when irrational optimists buy a stock, smart money sells, and when irrational pessimists sell a stock, smart money buys, thereby eliminating the effect of the irrational traders on market price. From a theoretical point of view, it is far from clear that smart money has the power to drive market prices to fundamental values. For example, in one model with both feedback traders and smart money, the smart money tended to amplify, rather than diminish, the effect of feedback traders, by buying in ahead of the feedback traders in anticipation of price increases they would cause (Shiller 2003). In addition to search costs, investors might choose mutual funds with high expenses if high-expense funds provided better service than other funds.

Barber et al. (2005) asserted that different levels of service are unlikely to explain their results, since first-rate service and low expenses are not mutually exclusive. For example, Vanguard, which is well-known for its low-cost mutual fund offerings, has won numerous service awards. Barber and Odean (2003) concluded that either models of optimal asset location are incomplete or a substantial fraction of investors are misallocating their assets. Though tax considerations leave clear footprints in the data they analyzed, many households could improve their after-tax performance by fully exploiting the tax-avoidance strategies available on equities.

### 3.9.5. The Media

Media may well have an important role in directing this public attention toward markets, which may consequently result in abnormal market behavior. Stock-market price increases generate news stories, which generate further stories about new-era theories that explain the price increases, which, in turn, generate more news stories about the price increases (Shiller 2002). In the United Kingdom, Diacon (2004) found that lay investors perceive higher risks in investing in financial services products than do their financial advisers (coupled with an inherent optimism about likely benefits) has substantial ramifications in the light of recent reports, such as the "Sandler Review". This may have the effect of encouraging consumers to deal directly with providers rather than via independent financial advisers.

Dispensing with the services of financial advisers is likely to lead consumers to make more conservative investment choices: for example, by investing too little in equities and too much in fixed-income assets when saving retirement. As a result, consumers may find themselves with surprisingly inadequate levels of savings to meet future commitments such as a pension on retirement. Hon (2013b) studied the investment attitude and behavior of the small investors on derivatives markets in Hong Kong. He found that the most decisive factor that could influence small investor's decision making is highly accessible and updated. In total, 37.8% and 25.8% of respondents considered the Internet and news/magazines/newspaper, respectively, as the decisive factor.

### 3.9.6. Emotions and Sentiments

There are serious questions concerning whether the phenomenon on excess volatility exists in the first place and, and if it does, whether abandonment of assumptions of rational expectations in favor of assumptions of mass psychology and fads as primary determinants of price changes is the best avenue for current research (Kleidon 1986). Using common sense, one knows that the stock market could repeat the performance of recent years. That possibility seems quite real, just as real as the possibility of a major correction in the market. The question is how the private investor feels when he fills out his choice of mutual funds for his retirement scheme. How this person feels depends on his experiences in investing.

If one has been out of the market without participating in earning money that other investors may have done, one may be feeling a sharp pain of regret. Regret has been found by psychologists to be one of the strongest motivations to make a change in something. Envy is another dominant characteristic: To see other people having made more money in the stock market than oneself has made from work is a painful experience, especially since it diminishes one's ego. In case the other people were smarter, one feels like a fool, and even if they were not any smarter, just lucky, it may not feel any better.

A common feeling in this situation is that if one can participate just one more year in rising stock market everything will be much better and mitigate the pain. One may also think that the potential loss will be much more diminishing to one's ego than the failure to participate has already been. One may also realize that one takes the risk of entering the market just as it begins a downward turn. However, the psychological cost of such a potential future loss may not be so much greater relative to the very real regret of having been out of the market in the past.

Barberis et al. (1998) presented a parsimonious model of investor sentiment, or of how investors form expectations of future earnings. The model they proposed was motivated by a variety of psychological evidence; in making forecasts, people pay too much attention to the strength of the evidence they are presented with and too little attention to its statistical weight. Loewenstein et al. (2001) proposed an alternative theoretical perspective, the risk-as-feelings hypothesis, which highlights the role of affect experienced at the precise moment of decision-making. Drawing on research from clinical, physiological, and other subfield of psychology, they showed that emotional reactions to risky situations often diverge from cognitive assessments of those risks. When such divergence occurs, emotional reactions often drive behavior. The risk-as-feelings hypothesis is shown to explain a wide range of phenomena that have resisted interpretation in cognitive–consequentialist terms.

If one participates in the market today for a while and ponders whether get out or not, he has a fundamentally different emotional frame of mind. This person feels satisfaction and probably pride in his past successes, and he will certainly feel wealthier. One may feel as gamblers do after they have "hit big-time", i.e., that one is gambling with the "house money", and therefore has nothing to lose emotionally by wagering again. The concept of gambling with the house money is a theory about people's risk preferences and is related to mental accounting. Investors will generally become more risk-averse in the case of prior losses and less risk-averse in the case of prior gain (Barberis and Thaler 2003); they will also take greater risks as their profits grow.

This provides support for the notion that successful traders are more likely to be overconfident. The emotional state of investors when they decide on their investment is no doubt one of the most important factors causing bull market. From the neuroscience literature, Peterson (2002) demonstrated correlations between reward anticipation and the arousal of affect (feelings, emotions, moods, attitudes, and preferences). He briefly outlined an investment strategy for exploiting the event-related security-price pattern described by the trading strategy "buy on the rumor and sell on the news".

In their research, Chow et al. (2015) conducted a survey to examine whether the theory developed in Lam et al. (2010, 2012) and Guo et al. (2017a) holds empirically, by studying the behavior of different types of Hong Kong small investors in their investment, especially during financial crisis. They found that determinants of the Hong Kong small investors' investment decision have uniform views as to the

ascending order of importance of time horizon, sentiment, and risk tolerance. Time horizon is the least important factor, and risk tolerance is the most important factor.

## 3.10. Volume and Volatility

Fong and Wong (2007) applied the volatility–volume regressions to the daily realized volatility of common stocks to study sources of volatility predictability. They found that unexpected volume can explain half of the variations in realized volatility and that the ARCH effect is robust in the presence of volume.

Xiao et al. (2009) studied the relationship between volume and volatility in the entire Australian Stock Market for different firm size and trading volume. They found that daily trading volume has significant explanatory power on the variance of daily returns. Actively traded stocks having a larger number of information arrivals per day will have a larger impact of volume on the variance of daily returns. Low trading volume and small firm lead to a higher persistence of GARCH effects, unlike the elimination effect for the top most active stocks. In general, the elimination of both ARCH and GARCH effects by introducing the volume variable on all other stocks, on average, is not as much as that for the top most active stocks. The elimination of both ARCH and GARCH effects by introducing the volume variable is higher for stocks in the largest volume and/or the largest market capitalization quartile group. Their empirical findings rejected the pure random-walk hypothesis for stock returns, and they concluded that the relationship between volume and volatility is not a statistical fluke. Unlike most anomalies, the volume effect on volatility is not likely to be eliminated after its discovery.

## 3.11. Trading Rules and Technical Analysis

If investors could make significant profit when they use any tool in technical analysis, adopt any trading rule, or employ any indicator in their investment, then we will consider this is an anomaly because this shows that the market is not efficient so that investors could have opportunity to make profit. There are many studies in this area. We list a few here.

We first discuss the adaptation of indicators and trading rules. For instance, Wong et al. (2001) introduced a new stock market indicator by using both E/P ratios and bond yields, and developed two statistics to test the following hypotheses:

**Hypothesis 1 (H1).** *Using the proposed indicator could make significantly good profit from the markets.*

**Hypothesis 2 (H2).** *Using the proposed indicator could beat the buy-and-hold (BH) strategy.*

In order to test the hypotheses, they examined the performance of their proposed indicator in five different stock markets, namely the UK, USA, Japan, Germany, and Singapore stock markets. Firstly, from their empirical study, they did not reject the hypothesis H1 in (i) and concluded that their indicator could produce buy and sell signals that investors could escape from most, if not all, of the major crashes, catch many of the bull markets, and generate significantly good profit.

Thereafter, they conducted an analysis to test the hypothesis H2 in (ii), and their analysis led them not to reject the hypothesis in (ii) and to conclude that their proposed indicator performs better than the BH strategy, because using their proposed indicator enabled investors to make significant higher profit then the BH strategy.

McAleer et al. (2016) developed some new indicators, or new trading rules, that can profiteer from any main financial crisis, and they examined the applicability of their proposed indicators/trading rules on the 1997 Asian Financial Crisis (AFC), the 2000 dot-com crisis (DCC), and the 2007 Global Financial Crisis (GFC). They examined the two hypotheses H1 and H2 in (i) and (ii) with their proposed indicators.

The empirical study did not reject (i) and concluded that using the signals generated by their proposed indicators/trading rules generate significantly good profit from the markets during AFC,

DCC, and GFC. Their empirical study also did not reject (ii), and they concluded that their proposed indicators/trading rules beat the BH strategy, because by using their proposed indicators/trading rules, investors could make very huge profit from the markets during AFC, DCC, and GFC; nonetheless, by adopting the BH strategy, all the profits are eaten up by the downtrend of the crisis and investors end up not having any profit or even bear big loss.

Chong et al. (2017) developed a new market sentiment index by using HIBOR, short-selling volume, money flow, the turnover ratio, the US and Japanese stock indices, and the Shanghai and Shenzhen Composite indices. Thereafter, they used the index as a threshold variable to determine different states in the market and applied the threshold regression to generate buy-and-sell signals. They illustrated the applicability of their proposed trading rules on the HSI or S&P/HKEx LargeCapIndex by testing the two hypotheses H1 and H2 in (i) and (ii), with the now-proposed indicator as their proposed approach.

Their empirical study did not reject (i), and they concluded that the use of their proposed approach generated significant profit from the Hong Kong market; the study did not reject (ii), and it concluded that their proposed approaches beat the BH strategy when investors buy the stock indices when the sentiment index is smaller than the lower threshold value, and vice versa.

Using both intraday and daily data, Lam et al. (2007) examined both surges and plummets of stock price and construct momentums and five trading rules of trading in stocks. They found that all their proposed trading rules cannot get any significant profit in both European and American stock markets but can get significant profit from the Asian stock markets. Their findings accept market for European and American stock markets but reject efficiency in the Asian stock markets, implying that the Asian stock markets are not as efficient as American and European stock markets.

There are many trading rules. An easy one is the single lump-sum investment rule (LS) that one invests all the fund at the beginning. Another popular one is the dollar-cost averaging investment rule (DCA) in which, regardless of ups and downs in the markets, one invests a fixed amount of money periodically over a given time interval in equal installments. This approach could avoid risk and the devastating effect when the market crashes suddenly. The literature shows that the LS rule outperforms the DCA rule when the market is uptrend, and the DCA rule outperforms the LS rule when the market is downtrend or mean-reverted.

Does the LS rule really outperform the DCA rule when the market is uptrend? Lu et al. (2020) conjectured that the DCA rule could still outperform the LS rule when the market is uptrend. To show that their conjecture could hold true, they applied both Sharpe Ratio (SR) and economic performance measure (EPM) and compared the performance of both LS and DCA rules in both accumulative and disaccumulative situations. They showed that, when the trend is not too upward, the DCA rule performs better than the LS rule in almost all the situations. In addition, when the market is uptrend, the DCA rule could still outperforms the LS rule in many situations, especially when volatility is high and when longer investment horizon is chosen.

Thus, the authors concluded that their conjecture hold true that the DCA rule could outperform the LS rule in many situations even in the situation the market is in the uptrend. Together with the findings in the literature that the DCA rule outperforms the LS rule when the market is downtrend or mean-reverted, the authors recommended that investors not choose the LS rule but use the DCA rule in their investment.

We note that one could apply the rules in the above papers to make good profit in their studying periods. If this is the case, then we will consider this is an anomaly, because this shows that the market is not efficient so that investors could have opportunity to make profit. However, there is a chance that the rules may not be able to make money after the rules released. If this is the case, then the market is still efficient and the anomaly disappears. Now, we turn to discuss using technical analysis (TA) to generate profit.

There are many studies that show that technical analysis can be used to generate profit. For example, to examine whether TA is profitable, Wong et al. (2003) used two popular technical tools—moving average (MA) and relative strength index (RSI)—and introduced two statistics to test

the two hypotheses, H1 and H2, in (i) and (ii), with the now-proposed indicators MA and RSI. Using the data from the Singapore stock market, they accepted both H1 and H2 and concluded that both MA and RSI could be used to make significantly positive profit and both MA and RSI beat the BH strategy significantly.

In addition, to examine whether TA is profitable, Wong et al. (2005) examined the performance of different MAs in the Taiwan, Shanghai, and Hong Kong stock markets and tested the hypotheses H1 and H2 in (i) and (ii) with the now-proposed indicators, are MA rules from MA family, by using the Greater China data. Their empirical findings did not reject H1, and they concluded that, in general, all MAs from the entire MA family can generate significantly positive profit and accept H2, and conclude that, in general, all MAs from the entire MA family generate significantly higher profit than the BH strategy in the two subperiods before and after the 1997 AFC, and in the entire period, as well as in all the bull, bear, and mixed markets.

Moreover, they conducted a wealth analysis and examined how much more wealth one can get by using all the MA rules from the entire MA family and concluded that different MA rules could yield different cumulative wealth, which could be as much as hundreds of times more than that obtained by choosing the BH strategy, when transaction costs have been considered. Without considering transaction costs, the cumulative wealth is much higher. Their findings and observations imply that the MA family is useful in investment that can create significant higher wealth so that we can reject market efficiency in the Greater China markets.

Like many other studies in TA rules, the above studies conclude that the TA rules are useful and can generate higher profit so that we can consider this is an anomaly. Nonetheless, not all studies make the same conclusions. Some could conclude that TA rules are not useful or at least not useful in some periods or in some markets. For example, to test the hypotheses H1 and H2 in (i) and (ii), Kung and Wong (2009a) made the following conjecture:

**Conjecture 1.** *Using TA rules may not be able to generate significant profit recently and the anomaly is disappearing.*

In order to test whether their conjecture holds true, they applied three most commonly used MA rules and tested whether using these three MAs rules could enable investors to make a significant profit in all the periods they studied in the Singapore stock market. From their empirical study, they did accept H1 in (i) and concluded that using the three MAs did generate significantly higher profit in the 1988–1996 period, but they reject H1 in (i), and they concluded that the three MAs did not generate significantly higher profit in the 1999–2007 periods.

In addition, their analysis led them to accept H2 in (ii) and conclude that using the three MAs did generate significantly higher profit than adopting the BH strategy in the 1988–1996 period, but they rejected H2 in (ii) and concluded that the three MAs did not generate significantly higher profit than adopting the BH strategy in the 1999–2007 periods. Based on their findings, market efficiency was rejected before the 1997 AFC, but was rejected for the period after the 1997 AFC in the Singapore stock market. This could mean that the anomaly is disappearing after the trading rules being introduced.

In addition, to test the hypotheses H1 and H2 in (i) and (ii) and test whether Conjecture 1 holds true, Kung and Wong (2009b) conducted a similar analysis in the Taiwan market. From their analysis, H1 in (i) was strongly accepted, and they concluded that the two popular TA rules could be used to generate significant profit in the 1983–1990 period. However, H1 in (i) was not so strongly accepted in the 1991–1997 period, and they concluded that the two popular TA rules could be used to generate only marginally significant profit in the 1991–1997 period.

Moreover, H1 in (i) was strongly rejected in the 1998–2005 period, and they concluded that the two popular TA rules could not be used to generate any significant profit in the 1998–2005 period. Based on their findings, market efficiency was strongly rejected in the 1983–1990 period, weakly rejected in the 1991–1997 period, and strongly accepted in the 1985–2005 period for the Taiwan stock market. Thus, the empirical findings from Kung and Wong (2009a, 2009b) support their conjecture that using TA

rules could be useful in the past, but it may not be able to generate significant profit recently, and the anomaly is disappearing. Readers may refer to Chan et al. (2014) for further information.

## 4. Behavioral Finance

There are many different areas in Behavioral Finance, including the topics discussed in the next section. Readers may refer to Wagner and Wong (2019) and the references therein for more information. Here, we only discuss a few.

### 4.1. Behavioral Finance and Market Efficiency

Behavioral Finance is a new approach to financial markets that has emerged, at least in part, in response to the difficulties faced by the traditional paradigm. In broad terms, it argues that some financial phenomena can be better understood using models in which some agents are not fully rational. More specifically, it analyzes what happens when we relax one, or both, of the two tenets that underlie individual rationality (Barberis and Thaler 2003). Behavioral Finance is the study of the influence of psychology on the behavior of financial practitioners and the subsequent effect on markets. In any situation that causes market inefficiency, as long as there exist sufficient explanations that can help to explain any of the anomalies or there is any way to maintain the relationship between information and stock price, it is the weak-form market efficiency, at least (Mullainathan et al. 2005). If Behavioral Finance makes it, then Behavioral Finance supports EMH. Since Behavioral Finance studies the behavior of investors and helps explain that market anomalies are caused by investors, it means that Behavioral Finance supports EMH when the below three assumptions of Fama (1965a) might not hold:

**Assumption 1 (A1).** *Rational investor.*

**Assumption 2 (A2).** *Independent deviation from rationality.*

**Assumption 3 (A3).** *Arbitrage.*

As Fama (1965a, 1970) claimed that any one of above three assumptions holds will make EMH effective, and A1 is stricter that A2, and meanwhile A2 is stricter than A3, Behavioral Finance explains why A1 or A2 or A3 hold does not hold.

### 4.2. Overconfidence

The key behavioral factor and perhaps the most robust finding in the psychology of judgment needed to explain A1 or A2 or A3 is overconfidence. Overconfidence is sometimes reversed for very easy items. Overconfidence implies over-optimism about the individual's ability to succeed in his endeavors (Frank 1935). Such optimism has been found in a number of different settings. Men tend to be more overconfident than woman, though the size of difference depends on whether the task is perceived to be masculine or feminine (Hirshleifer 2001). Economists have long asked whether investors who misperceive asset returns can survive in a competitive asset market such as a stock or a currency market.

De Long et al. (1991) concluded that there is, in fact, a presumption that overconfident investors—even grossly overconfident investors—will tend to control a higher proportion of the wealth invested in securities markets as time passes. This presumption is based on the empirical observations that (a) most investors appears to be more risk-averse than log utility; and (b) idiosyncratic risk is large relative to systematic risk. Under these conditions, investors who are mistaken about the precision of their estimate of the returns expected from a particular stock will end up taking on more systematic risk. Taken as a group, these investors will exhibit faster rates of wealth accumulation than fully rational investors with risk aversion greater than given by log utility.

Kyle and Wang (1997) showed that overconfidence may strictly dominate rationality since an overconfident trader may not only generate higher expected profit and utility than his rational opponent,

but also higher if he is also rational. This occurs because overconfidence acts like a commitment device in a standard Cournot duopoly. As a result, for some parameter values the Nash equilibrium of two-fund game is a Prisoner's Dilemma in which both funds hire overconfident managers. Thus, overconfidence can persist and survive in the long run.

Daniel et al. (1998) developed a theory based on investor overconfidence and on changes in confidence resulting from biased self-attribution of investment outcomes. The theory implies that investors will overreact to private information signals and underreact to public information signals. Odean (1998b) finds that people are overconfident. His paper examines markets in which price-taking traders, a strategic-trading insider, and risk-averse market-makers are overconfident. Overconfidence increases expected trading volume, increases market depth, and decreases the expected utility of overconfident traders.

Benos (1998) studied an extreme form of posterior overconfidence where some risk-neutral investors overestimate the precision of their private information. The participation of overconfident traders in the market leads to higher transaction volume, larger depth, and more volatile and more information prices. An important anomaly in finance is the magnitude of volume in the market. For example, Odean (1999) noted that the annual turnover rate of shares on the New York Stock exchange is greater than 75 percent, and the daily trading volume of foreign-exchange transactions in all currencies (including forwards, swaps, and spot transactions) is equal to about one-quarter of the total annual world trade and investment flow. Odean (1999) then presented data on individual trading behavior, suggesting that extremely high volume may be driven, in part, by overconfidence on the part of investors.

Individual investors who hold common stocks directly pay a tremendous performance penalty for active trading. Of 66,465 households with accounts at a large discount broker during 1991 to 1996, those that trade most earn an annual return of 11.4 percent, while the market returns 17.9 percent. The average household earns an annual return of 16.4 percent, tilts its common stock investment toward high-beta, small-value stocks, and turns over 75 percent of its portfolio annually. Overconfidence can explain high trading levels and the resulting poor performance of individual investors (Barber and Odean 2000a).

Barber and Odean (2000b) reported their analysis, using account data from a large discount brokerage firm, of the common stock investment performance of 166 investment clubs from February 1991 through January 1997. The average club tilts its common stock investment toward high-beta, small-cap growth stocks and turns over 65 percent of its portfolio annually. The average club lags the performance of a broad-based market index and the performance of investors. Moreover, 60 percent of the clubs underperform the index.

Gervais and Odean (2001) developed a multi-period market model describing both the process by which traders learn about their ability and how a bias in this learning can create overconfident traders. A trader's expected level of overconfidence increases in the early stages of his career. Then, with more experience, he comes to better recognize his own ability. The patterns in trading volume, expected profits, price volatility, and expected prices resulting from this endogenous overconfidence are analyzed. Theoretical models predict that overconfident investors trade excessively.

Barber and Odean (2001a) tested this prediction by partitioning investors on gender. Psychological research demonstrates that, in areas such as finance, men are more overconfident than women. Thus, theory predicts that men will trade more excessively than women. They documented that men trade 45 percent more than women. Trading reduces men's net return by 2.65 percentage points a year, as opposed to 1.72 percentage points for women. People (especially males) seem to trade too aggressively, incurring higher transactions costs, without higher return. From the view that the behavior of overconfident investors is irrational, and the anomaly arises because investors are not rational, Behavioral Finance does not confront, but supports EMH. Once A1 or A2 or A3 is satisfied, the market is still efficient.

### 4.2.1. Utility

One of the main reasons that EMH is rejected in many cases and there are many market anomalies in the market is that different investors could have different types of utilities.

### 4.2.2. Investors with Different Shapes in Their Utility Functions

Many scholars, for example, Bernoulli (1954), believe that investors are risk averse; that is, their utility is increasing concave. Many financial models are developed based on the foundational assumption that investors are risk averse or their utility is increasing concave. For example, Markowitz (1952a) developed the mean–variance (MV) portfolio optimization theory based on this assumption.

In reality, investors' utility may not be increasingly concave. It could be increasingly convex (that is, investors are risk-seeking) or S-shaped or reverse S-shaped. Tobin (1958), Hammond (1974), Stoyan (1983), Wong and Li (1999), Li and Wong (1999), Wong (2006, 2007), Wong and Ma (2008), Levy (2015), Bai et al. (2015), Guo and Wong (2016), and many others have built up their theories by assuming that investors could be risk averse or risk seeking.

Kahneman and Tversky (1979) suggested investors' utility[1] could be concave for gains and convex for losses, implying that investors have a S-shaped utility function. On the other hand, Thaler and Johnson (1990) observed that investors are more risk-seeking on gains and more risk-averse on losses, inferring that investors have a reverse S-shaped utility function. Other academics, for example, Levy and Wiener (1998), Levy and Levy (2002, 2004), Wong and Chan (2008), and Bai et al. (2011b) developed their theories based on the assumption that investors possess S-shaped or reverse S-shaped utility function.

### 4.2.3. Other Utility Functions

Academics not only use the shape of utility functions to measure the behaviors of different investors, but also use other forms of utility functions to measure their behaviors, for example, regret-aversion (Guo et al. 2015; Egozcue et al. 2015), disappointment-aversion (Guo et al. 2020), and many others. In addition, Guo et al. (2016) developed the exponential utility function with a 2n-order and established an estimation approach to find the smallest possible n to provide a good approximation for any integer n.

Chan et al. (2019a) proposed using polynomial utility functions to measure the behavior of risk-averters and risk-seekers. Wong and Qiao (2019) proposed including both risk-averse and risk-seeking components to measure the behavior of investors who could gamble and buy insurance together, or buy any less risky and more risky assets at the same time. Egozcue and Wong (2010a) propose a utility function for Segregation and Integration.

### 4.3. Portfolio Selection and Optimization

Portfolio optimization and portfolio selection are the founding theories of modern finance, and they are one of the major areas in Behavioral Finance. They are related to Behavioral Finance because different investors with different utilities could make different selections and get different optimizations. The foundational portfolio optimization theory developed by Markowitz (1952a), to find out how investors will choose their portfolios, requests assumption of risk-aversion on investors.

Nevertheless, the MV portfolio optimization theory developed by Markowitz (1952a) has been found to have serious problem in its (plug-in) estimation (Michaud 1989), while Bai et al. (2009a) not only prove that the serious estimation problem is natural and it is overestimation, not underestimation. In addition, they find out the magnitude of the overestimation. Thus, one is not surprised they can apply the asymptotic properties of eigenmatrices for large sample covariance matrices (Bai et al. 2011c)

---

[1]  Kahneman and Tversky (1979) and others call it value function, while we call it utility function.

to find out the estimation (they call it bootstrap-corrected estimation) that is consistent to the true optimal return.

Nonetheless, the problem of the bootstrap-corrected estimation is that it does not have a closed-form. To solve the problem, Leung et al. (2012) extended the theory by developing the estimation with closed-form, and Bai et al. (2009b) extended the theory of portfolio optimization for the problem of self-financing. In addition, Bai et al. (2016) further extended the model by employing the spectral distribution of the sample covariance to develop the spectral-corrected estimation that performs better than both plug-in and bootstrap-corrected estimations.

The problem of all the above estimations developed by Markowitz (1952a), Bai et al. (2009a, 2009b), Leung et al. (2012), and many others is that the estimations are the same for any investor with risk-averse utility. Nonetheless, it is well-known that different investors could choose different optimal portfolios. In addition, Guo et al. (2019b) established some properties on efficient frontiers and boundaries of portfolios by including background risk in the model and by using several approaches, including MV, mean–VaR, and mean–CVaR approaches.

Many studies have explored how to get solutions for different investors. For example, Li et al. (2018) applied the Maslow portfolio selection model (MPSM) to develop a model that could take care of the need of investors with low financial sustainability who will first look into their lower-level (safety) need, and thereafter look into their higher-level (self-actualization) need, to obtain their optimal return. They illustrated their model by comparing the out-of-sample performance of the traditional model and their proposed model by using the real American stock data. They observed that their proposed model outperformed the traditional model to get the best out-of-sample performance.

We note that one can modify the model developed by Li et al. (2018) to find the optimal return for investors with high financial sustainability who prefer to looking into their higher-level need first, and then satisfying their lower-level need. Though both investors with high and low financial sustainability are risk-averse, their choices are different in the portfolio selections.

There are many applications using the theory of portfolio optimization (for example, Abid et al. (2009, 2013, 2014), Hoang et al. (2015a, 2015b, 2018, 2019), Mroua et al. (2017), Bouri et al. (2018), and many others).

### 4.4. Stochastic Dominance

Stochastic dominance (SD) is one of the most important areas in Behavioral Finance, because SD can be used to compare the performance of different assets, which is equivalent is the preferences of investors with different utilities. We discuss some important SD papers in this section. Readers may refer to Levy (2015), Sriboonchitta et al. (2009), and the references therein for more information.

### 4.4.1. Stochastic Dominance for Risk-Averters and Risk-Seekers

SD is one of the most important selection rules for both risk-averters and risk-seekers. Quirk and Saposnik (1962), Fishburn (1964, 1974), Hadar and Russell (1969, 1971), Hanoch and Levy (1969), Whitmore (1970), Rothschild and Stiglitz (1970, 1971), Tesfatsion (1976), Meyer (1977), and others developed the SD rules for risk-averters. On the other hand, Hammond (1974), Meyer (1977), Hershey and Schoemaker (1980), Stoyan (1983), Myagkov and Plott (1997), Wong and Li (1999), Li and Wong (1999), Anderson (2004), Post and Levy (2005), Wong (2006, 2007), Levy (2015), Guo and Wong (2016), and others developed the SD rules for risk-seekers.

### 4.4.2. Stochastic Dominance for Investors with (Reverse) S-Shaped Utility Functions

Friedman and Savage (1948) observed that many individuals buy insurance, as well as lottery tickets, and it is well-known that utility with strictly concavity or strictly convexity cannot explain this phenomenon. To solve this problem, academics developed S-shaped and reverse S-shaped utility functions, while Levy and Levy (2002, 2004) and others developed the SD theory for investors with S-shaped and reverse S-shaped utility functions. We call the SD theory for individual with S-shaped

prospect SD (PSD) and the SD theory for individual with reverse S-shaped utility function Markowitz SD (MSD).

Wong and Chan (2008) extended the PSD and MSD theory to the first three orders. They developed several important properties for MSD and PSD. For example, they showed that the dominance of assets in terms of PSD (MSD) is equivalent to the expected utility preference of the assets for investors with (reverse) S-shaped utility function.

### 4.4.3. Almost Stochastic Dominance

The theory of almost stochastic dominance (ASD) was developed by Leshno and Levy (2002, LL) to measure a preference for "most" decision makers but not all decision makers. However, Tzeng et al. (2013, THS) found that expected-utility maximization does not hold in the second-degree ASD (ASSD) defined by LL. Thus, they suggested to use another definition of the ASSD definition that possesses the property. Nonetheless, Guo et al. (2013) proved that, though LL's ASSD does not satisfy the expected-utility maximization, it possesses the hierarchy property, whereas though THS's ASSD possesses the expected-utility maximization, it does not possess the hierarchy property.

Guo et al. (2014) extended the ASD theory by developing the necessary conditions for the ASD rules. In addition, they established several important properties for ASD. Guo et al. (2016) further extended the theory of ASD theory by including the theory of ASD for risk-seekers. In addition, they established some relationships between ASD for both risk-averters and risk-seekers. Tsetlin et al. (2015) established the theory of generalized ASD (GASD), and Guo et al. (2016) compared ASD and GASD and pointed out their advantages and disadvantages.

### 4.4.4. Stochastic Dominance Tests and Applications

There are many stochastic dominance tests that can be used in Behavioral Finance. The commonly used SD tests include Davidson and Duclos (2000), Linton et al. (2005), Linton et al. (2010), Bai et al. (2011b, 2015), and Ng et al. (2017). We note that Lean et al. (2008) conducted simulations to compare the performance of different SD tests and found that the SD test developed by Davidson and Duclos (2000) has decent size and power.

The SD tests developed by Bai et al. (2011b, 2015) are improvements of the SD test developed by Davidson and Duclos (2000). Thus, the SD tests developed by Bai et al. (2011b, 2015) also have decent size and power. Ng et al. (2017) conducted simulations and found that their proposed SD test also had decent size and power. Chan et al. (2019a) developed a third-order SD test.

There are many studies that have applied stochastic dominance tests to test for market efficiency and check whether there is any anomaly in the market (for example, Fong et al. (2005, 2008), Wong et al. (2006, 2008, 2018a), Lean et al. (2007, 2010a, 2010b, 2013, 2015), Gasbarro et al. (2007, 2012), Wong (2007), Chiang et al. (2008), Abid et al. (2009, 2013, 2014), Qiao et al. (2010, 2012, 2013), Chan et al. (2012), Qiao and Wong (2015), Hoang et al. (2015a, 2015b, 2018, 2019), Tsang et al. (2016), Mroua et al. (2017), Bouri et al. (2018), and Valenzuela et al. (2019), among others).

### 4.5. Risk Measures and Performance Measures

The theory of risk measures and performance measures is one of the most important theories in Behavioral Finance because it can be used to measure the preferences of investors from different types of utilities. We discuss some different risk measures in this section.

### 4.5.1. Mean Variance Rules

Markowitz (1952b) and Tobin (1958) first proposed the mean–variance (MV) selection rules for risk-averters, while Wong (2007) extended the theory by introducing the MV selection rules for risk-seekers and established the relationship between SD and MV rules for both risk-averters and risk-seekers. Chan et al. (2019b) further extended the theory by introducing the moment rules for both risk-averters and risk-seekers, and establishing the relationship between SD and the moment rules.

### 4.5.2. Sharpe Ratio

The Sharpe ratio, developed by Sharpe (1966), is one of the most commonly used reward-to-risk measures, but it does not provide the testing of the ratio. To circumvent the limitation, Lo and others developed the testing statistics for the Sharpe ratio. Leung and Wong (2008a) extended the theory further by establishing the testing statistic to multivariate settings and developing the asymptotic distribution of the statistic and related properties. Chow et al. (2019b) showed the relationship between SD and the Sharpe ratio. Wong et al. (2012) introduced the mixed Sharpe ratio to the theory and showed that the mixed Sharpe ratio changes over time.

### 4.5.3. Mean–Variance Ratio

It is well-known that the Sharpe ratio test is not uniformly most powerful unbiased (UMPU). To get a UMPU test, Bai et al. (2011d) established the mean–variance-ratio (MVR) statistic to test the equality of mean–variance ratios for different assets. Bai et al. (2012) further improved the test by removing the background risk. They showed that their proposed MVR statistic is UMPU in any sample, no matter whether the sample is big or small.

### 4.5.4. Omega Ratio

The Omega ratio introduced by Keating and Shadwick (2002) is one of the most important performance measures by using the probability-weighted ratio of gains and losses for any threshold return target. Guo et al. (2017b) developed the properties to study the relationships between the Omega ratio and (a) the first-order SD; (b) the second-order SD for risk-averters; and (c) the second-order SD for risk-seekers. Chow et al. (2019b) further established the necessary conditions between the Omega ratio and SD for both risk-averters and risk-seekers, and demonstrated that the Omega ratio outperforms the Sharpe ratio in many cases.

### 4.5.5. Economic Performance Measure

Homm and Pigorsch (2012) developed the economic performance measure (EPM), which is related to Behavioral Finance, because it is related to SD, which, in turn, is related Behavioral Finance. Thus, EPM is related to Behavioral Finance. To make the EPM become useful in the comparison of different assets, Niu et al. (2018) developed the theory of construction confidence intervals for EPMs, including one-sample and two-sample confidence intervals, and derived the asymptotic distributions for one-sample confidence interval and for two-sample confidence interval for samples that are independent. The testing approach developed by Niu et al. (2018) is robust for many dependent cases.

### 4.5.6. Other Risk Measures and Performance Measures

Because variance gives the same weight in measuring downside risk as well as upside profit for any prospect, it is not a good measure to capture the downside risk. To circumvent the limitation, several risk measures and performance measures have been proposed, including Value-at-Risk (VaR, Jorion 2000; Guo et al. 2019b), conditional-VaR (C-VaR, Rockafellar and Uryasev 2000; Guo et al. 2019b), Kappa ratio (Kaplan and Knowles 2004), Farinelli and Tibiletti (FT, Farinelli and Tibiletti 2008), economic performance measure (Homm and Pigorsch 2012), and others. In addition, Ma and Wong (2010) proved that VaR is related to first-order SD and C-VaR is related to second-order SD. Niu et al. (2017) proved that the Kappa ratio is related to SD, and Guo et al. (2019a) proved that the F–T ratio is related to SD for both risk-averse and risk-seeking investors under some conditions.

### 4.5.7. Applications of Risk Measures and Performance Measures

There are many applications of using risk measures and performance measures to test for market efficiency, and check whether there is any anomaly in the market. Examples include Sharpe (1966), Keating and Shadwick (2002), Kaplan and Knowles (2004), Broll et al. (2006, 2011, 2015),

Wong et al. (2008, 2018a), Leung and Wong (2008a), Fong et al. (2008), Abid et al. (2009, 2013, 2014), Qiao et al. (2010, 2012, 2013), Lean et al. (2010a, 2010b, 2013, 2015), Homm and Pigorsch (2012), Chan et al. (2012, 2019a, 2019b), Bai et al. (2013), Qiao and Wong (2015), Hoang et al. (2015a, 2015b, 2018, 2019), Tsang et al. (2016), Guo et al. (2017b, 2019a), Mroua et al. (2017), Niu et al. (2017), Bouri et al. (2018), and Chow et al. (2019b), among others.

### 4.5.8. Indifference Curves

The indifference curve, which was first developed by Tobin (1958), is one of the important areas in Behavioral Finance, because it reveals the behavior of both risk-averters and risk-seekers in the mean and variance diagram. Tobin (1958) proved that the indifference curve is increasingly convex for risk-averters, decreasingly convex for risk-seekers, averse (seeking) investors, and is horizontal for risk-neutral investors when assets follow the normal distribution. Meyer (1987) extended the theory by relaxing the assumption of normality and including the location-scale (LS) family to the theory of indifference curve.

Wong (2006, 2007) extended the theory to include the general conditions that were presented in Meyer (1987)Meyer Wong and Ma (2008) further extended the theory by introducing some general non-expected utility functions and the LS family with general n random seed sources, and established some important properties of the theory of indifference curves. To date, the literature only discusses indifference curves for risk-averters and risk-seekers. Broll et al. (2010) extended the theory by examining the behavior of indifference curves for investors with S-shaped utility functions.

### 4.6. Two-Moment Decision Models and Dynamic Models with Background Risk

The two-moment decision model is related to Behavioral Finance because it can be used to measure the behaviors of both risk-averters and risk-seekers. Many works have been done in this area. For instance, Alghalith et al. (2017) showed that the change of the price of expected energy will affect the demand for both energy and non-risky inputs, but the uncertain energy price only affects uncertain energy price but not the demands for the non-risky inputs for any risk-averse firm.

Alghalith et al. (2017) showed that the variance of energy price affects the demands of both non-risky inputs and energy decrease when the variance is vulnerable, but does not affect the demands of the non-risky inputs when there is only uncertain in energy price for any risk-averse firm. Guo et al. (2018a) contributed to the MV model with multiple additive risks by establishing some properties on the marginal rate of substitution between mean and variance. They also illustrated the properties by using the MV model with multiple additive risks to study banks' risk-taking behaviors.

In addition, Alghalith et al. (2016) extended the theory of the stochastic factor model with an additive background risk and the dynamic model with either additive or multiplicative background risks by including a general utility function in the models in which the risks are correlated with the factors in the models.

### 4.7. Diversification

Diversification is one of the important areas in Behavioral Finance. It is related to Behavioral Finance because different investors will have different behaviors in diversification. In the theory of portfolio selection developed by Markowitz (1952a), investors are assumed to be risk-averse, and there is only one best optimal portfolio that investors should choose. Li et al. (2018) showed that even investors are risk-averse, as they can choose different optimal portfolios if one looks into their safety need first, and thereafter, look into his/her self-actualization need while another one looks into his/her self-actualization need first, and then look into his/her safety need.

In addition, Li and Wong (1999) have shown that if all assets are independently and identically distributed, then it is true that risk-averters will choose the same optimal portfolio, which is the completely diversified portfolio. However, they found that investing in a single asset is the best choice for risk-seekers. Wong (2007) extended the theory for the diversification behaviors of both risk-averters

and risk-seekers to the gain, as well as to the loss, while Guo and Wong (2016) extended the theory further, in order to study the diversification behaviors of both risk-averters and risk-seekers in the multivariate settings.

To date, Li and Wong (1999), Wong (2007), and Guo and Wong (2016) only developed the diversification theory to study the diversification behaviors of both risk-averters and risk-seekers, to compare any pair of asset/portfolio(s) among a single asset, partially diversified portfolios, and completely diversified portfolio, but they have not developed any result in the comparison between any two portfolios of partial diversification. To circumvent the limitation, Egozcue and Wong (2010b) established a diversification theory to compare any pair of partially diversified portfolios, including completely diversified portfolios and individual asset.

Using the out-of-sample performance tool, DeMiguel et al. (2009) showed that the naive 1/N portfolio outperforms the "optimal" portfolio from the 14 models in terms of several commonly used measures in their study, and thus, they drew conclusion that the "optimal" portfolio is not optimal. Hoang et al. (2015b) found that risk-averters agree with Markowtiz to select the portfolios from the efficient frontier, while risk-seekers agree with De Miguel, Garlappi, and Uppal to select the equal-weighted portfolio. On the other hand, Bouri et al. (2018) found that risk-averters prefer the portfolios from the efficient frontier for low-risk with-wine portfolios but are indifferent between the portfolios from the efficient frontier and the naïve portfolio for any high-risk with-wine portfolios.

Statman (2004) showed that investors do not follow Markowtiz's suggestion to invest in the completely diversified portfolio and do not buy only one stock but buy a few. This is the well-known diversification puzzle. To provide a possible solution to the puzzle, Lozza et al. (2018) showed that investors' choices on optimal assets are similar if their utility are not too different.

Egozcue et al. (2011a) bridged the gap in the literature to develop some properties for the diversification behaviors for investors with reverse S-shaped utility functions that have never studied before. They found that the diversification preference for investors with reverse S-shaped utility functions are complicated and depend on the sensitivies toward losses and gains.

*4.8. Behavioral Models*

In this section, we discuss several behavioral models for Behavioral Finance that relate to market efficiency and anomalies in stock markets. So far, all the models discussed in Sections 4.1–4.7 are behavioral models. Thus, in this section, we discuss the behavioral models that are not discussed in Sections 4.1–4.7.

4.8.1. Behavioral Models for Some Financial Anomalies

By applying the concept of both conservatism and representativeness heuristics, Barberis et al. (1998) and others developed the Bayesian models, which can be used to explain investors' behavioral biases. Lam et al. (2010) extended their work by introducing a pseudo-Bayesian approach to reflect the biases from investors' behavior on each of each dividend being assigned (Thompson and Wong 1991, 1996; Wong and Chan 2004). Their model can be used to explain excess volatility, long-run overreaction, short-run underreaction, and their magnitude effect. Lam et al. (2012) improved the theory by establishing some new properties by using the pseudo-Bayesian model to explain the market anomalies and the investors' behavioral biases.

Fung et al. (2011) further improved the theory by considering the impact of a financial crisis. Guo et al. (2017a) improved the theory by first relaxing the normality assumption to any exponential family distribution for the earning shock of an asset that follows a random-walk model, with and without drift. They established additional properties to explain excess volatility, long-term overreaction, short-term underreaction, and their magnitude effects during financial crises, as well as the period of recovery thereafter.

The theory developed by Guo et al. (2017a) and the references therein can only explain some market anomalies, like excess volatility, long-run overreaction, short-run underreaction, and their

magnitude effect, but cannot be used to test it empirically. To circumvent the limitation, Fabozzi et al. (2013) developed several statistics that can be used to test whether there is any long-run overreaction and short-run underreaction, and their magnitude effect in the markets. They applied their statistics empirically and concluded that long-run overreaction, short-run underreaction, and their magnitude effect did exist in the markets they studied.

In addition to conducting statistical analysis to real data of stock prices to test whether there is any market anomaly, scholars could also use questionnaires to conduct surveys to examine investors' attitude on the market anomaly. For example, Wong et al. (2018b) distributed their questionnaires to small investors in Hong Kong, to conduct a survey to examine the behavior of investor behavior on long-term overreaction, short-term underreaction, and their magnitude effect. Their empirical findings support the theory developed by Guo et al. (2017a), and the references therein, that small investors in Hong Kong have both conservative and representative heuristics and they do use momentum and contrarian strategies in their investment.

### 4.8.2. Other Behavioral Models

The regret-aversion model is an important model for Behavioral Finance; for example, it can be used for investors to make decision in their portfolio investment (Barberis et al. 2006; Muermann et al. 2006). It can be used in many other areas, as well, for example, options (Sarver 2008) and hedging (Egozcue et al. 2015; Guo et al. 2015; Guo and Wong 2019).

On the other hand, the disappointment-aversion model developed by Bell (1982) and Loomes and Sugden (1982) can also be used in Behavioral Finance. For example, it can be used to determine the weights investors should hold stock and bond. Readers may refer to Guo et al. (2020) and the reference therein for more information.

Wan and Wong (2001) developed a behavioral model with incomplete information that can be used in refinancing during finance crisis. They found out the conditions to make financial crisis happen from one country to another one. In addition, Fry et al. (2010) and Fry-McKibbin and Hsiao (2018) developed statistics to test for contagion effect.

Given the studies in Sections 3.1 and 3.2, Fama (1998) claims that EMH survives in the abnormal returns brought by the Winner–Loser Effect and Momentum Effect. In particular, Fama insists that anomalies are chance results, which is consistent with the EMH, as it is obvious that overreaction to information is as common as underreaction to information. Moreover, the duration of abnormal returns before and after events is similar to the frequency of reversal of past events. Most importantly, consistent with market efficiency forecasts, the obvious anomalies may be due to different methodologies, as most long-term earnings anomalies tend to disappear as technology changes reasonably.

Some of the other literature has attempted to explain the above anomalies through the Behavioral Finance perspective, and the authors developed models. The first one is BSV model. Barberis et al. (1998) consider that there are two wrong paradigms when people make investment decisions: representative bias and conservative bias. The former refers to investors paying too much attention to the change patterns of recent data, but not enough attention to the overall characteristics of these data. The latter describes how investors cannot modify the increased forecasting model in time according to the changed situation. These two biases lead to underreaction and overreaction, separately. The BSV model explains how investors' decision-making models lead to market price changes deviating from the EMH.

By importing investor overconfidence and biased self-attribution, another two well-known psychological biases, Daniel et al. (1998) set up the DHS model. In the DHS model, overconfident investors are believed to overestimate their own prediction ability, underestimate their own prediction errors, over-trust private information, and underestimate the value of public information, which makes the weight of private signals in the eyes of overconfident investors higher than previous information and causes overreaction. While noisy public information can partially correct price inefficiencies when it arrives, overreacting prices tend to reverse when additional public information is available.

The third model to fix momentum anomalies is the HS model. The HS model, also known as the unified theoretical model, differs from the BSV model and DHS model in that it focuses on the mechanism of different actors rather than the cognitive bias of the actors (Hong and Stein 1999). The model divides participants into two categories: observers and momentum traders. Observers are assumed to make predictions based on future value information, while momentum traders rely entirely on past price changes.

Under the above assumptions, the model unifies the underreaction and overreaction as the basis. The model argues that the tendency of observers to underreact to private information first leads momentum traders to try to exploit this by hedging strategies, which in turn leads to overreaction at the other extreme.

### 4.9. Unit Root, Cointegration, Causality, and Nonlinearity

Unit root test, cointegration, and causality are important areas in Behavioral Finance because they can measure many different behaviors in Behavioral Finance. For example, Lam et al. (2006) developed three test statistics that can be used to test whether a series follows a random-walk or a stationary general-mean-reversed (GMR) model. It is investors' behavior to make stock prices follow a stationary general-mean-reversed (GMR) model. If the market is efficient, the stock price should follow a random-walk model. It is also because of investors' behavior that some stocks are moving together (cointegration) or not moving together, or one stock price could cause (causality) another one.

The authors have developed some unit root (Tiku and Wong 1998), cointegration (Penm et al. 2003; Wong et al. 2007), causality (Bai et al. 2010, 2011a, 2018), and nonlinearity (Hui et al. 2017) tests that related to Behavioral Finance.

There are many applications of unit root, cointegration, causality, and nonlinearity tests in many different areas of Behavioral Finance, including Manzur et al. (1999), Wong et al. (2004a, 2004b, 2007), Lam et al. (2006), Qiao et al. (2008a, 2008b, 2008c, 2009, 2011), Zheng et al. (2009), Chiang et al. (2009), Liew et al. (2010), Owyong et al. (2015), Vieito et al. (2015), Chow et al. (2018a, 2018b, 2019a), Batmunkh et al. (2018), Demirer et al. (2019), Gupta et al. (2019b), Zhu et al. (2019), Cheng et al. (2019), Lv et al. (2019), and many others.

### 4.10. Covariance and Copulas

Covariance and copulas are important areas in Behavioral Finance because they can measure the time-varying covariances that are caused by investors' different behavior over time; for example, Lam et al. (2010, 2012), Fung et al. (2011), Guo et al. (2017a), and many others showed that investor conservatism and representativeness heuristics cause excess volatility in financial markets.

We have developed some theories in covariance and copulas (see, for example, Egozcue et al. (2009, 2010, 2011a, 2011b, 2011c, 2012, 2013), Bai et al. (2009a, 2009b, 2011c), and Ly et al. (2019a, 2019b)). We have also conducted some analysis by using covariance and copulas (Tang et al. 2014).

### 4.11. Robust Estimation and Other Econometric Models/Tests

Robust estimation and other econometric models/tests are an important area in Behavioral Finance because they can measure the models in Behavioral Finance better. For example, Wong and Bian (2000) found that the robust Bayesian estimation introduced by Bian and Dickey (1996) could lead to mean square error (MSE) being greater than one thousand times smaller than that of the traditional least squares (LS) estimates when the error terms follow very heavy tails that are common in Behavioral Finance.

We have developed some theories in robust estimation and other econometric models/tests (see, for example, Wong and Miller (1990); Matsumura et al. (1990); Tiku et al. (1999a, 1999b, 2000); Wong and Bian (2005); Wong et al. (2001); Leung and Wong (2008b); Bian et al. (2011); and many others).

We have also used robust estimation to conduct many applications (see, for example, Wong and Bian (2000); Phang et al. (1996); Phang and Wong (1997); Wong et al. (2001); Fong and Wong (2006);

Qiao et al. (2008c); Bian et al. (2011); Raza et al. (2016); Xu et al. (2017); Chan et al. (2018); Guo et al. (2018b); Tsendsuren et al. (2018); Gupta et al. (2019a); Pham et al. (2020); and many others).

Fong and Wong (2007) applied the volatility–volume regressions to the daily realized volatility of common stocks to study sources of volatility predictability. They found that unexpected volume can explain half of the variations in realized volatility and find that the ARCH effect is robust in the presence of volume.

*4.12. Anchoring and Adjustment*

In many situations, people make estimates by starting from an initial value that is adjusted to yield the final answer. The initial value, or starting point, may be suggested by the formulation of the problem, or it may be the result of a partial computation. In either case, adjustments are typically insufficient. That is, different starting points yield different estimates, which are biased toward the initial values. We call this phenomenon anchoring (Tversky and Kahneman 1974).

Thus, anchoring refers to the decision-making process where quantitative assessments are required and where these assessments may be influenced by suggestions. People have in their minds some reference points (anchors), for example, in the study of Momentum Effect, anchors are previous stock prices. When they get new information, they adjust this past reference insufficiently (under-reaction) to new information acquired. Anchoring describes how individuals tend to focus on recent behavior and give less weight to longer time trends.

Anchoring can cause investors to underreact to new information (Fuller 1998). Values in speculative markets, like the stock market, are inherently ambiguous. It is hard to tell what the value of the Hang Seng Index should be. There is no agreed-upon economic theory that would provide an answer to this question. In the absence of any better information, past prices are likely to be important determinants of prices today. Therefore, the anchor, being the most recently remembered prices, causes the Momentum Effect.

## 5. Conclusions

Scholars could use data from stock markets all over the world to check whether the markets are efficient, as well as find whether there is any market anomaly. When there is any anomaly being discovered, scholars first confirm the existence of the market anomaly and thereafter look for any existing model to explain the anomaly. If scholars cannot estimate, evaluate, and forecast any model to explain the anomaly, scholars will then explain the anomaly by using quantitative analysis, modeling, or even building up a new theory to explain the anomaly that built up the theory of Behavioral Finance. However, if there is any unexplained anomaly, one may grasp the methods to profiteer by using the anomaly. On the one hand, this is a good way to offer investors valuable investment advice. On the other hand, in the long run, these anomalies may disappear unconsciously.

Many studies, for example, Frankfurter and Mcgoun (2000), argue that numerous empirical researches are not consistent with the EMH, and they conclude that debate on Behavioral Finance is not rigorous enough. In this paper, we revisited the issue on market efficiency and market anomalies. We first gave a brief review on market efficiency, including discussing some theories for market efficiency and reviewing some important works in market efficiency. We then reviewed different market anomalies, including Winner–Loser Effect, reversal effect, Momentum Effect, calendar anomalies that include January effect, weekend effect and reverse weekend effect, book-to-market effect, value anomaly, size effect, Disposition Effect, Equity Premium Puzzle, herd effect and ostrich effect, bubbles, and different trading rules and technical analysis.

Thereafter, we reviewed different theories of Behavioral Finance that could be used to explain market anomalies. Although we have discussed many studies on market efficiency and anomalies, there are still many theoretical contributions in other areas that could also be useful to explain and interpret market efficiency and anomalies. Readers may refer to Chang et al. (2016a, 2016b, 2016c, 2017, 2018) for contributions in other cognate areas that might be useful in theory and practice that

related to market efficiency and anomalies. Finally, we note that this review is useful to academics for their studies in EMH, anomalies, and Behavioral Finance; useful to investors for their decisions on their investment; and useful to policy makers in reviewing their policies in stock markets.

**Author Contributions:** Conceptualization, K.-Y.W., C.M., and W.-K.W.; writing—original draft preparation, K.-Y.W., C.M., and W.-K.W.; writing—review and editing, K.-Y.W., C.M., M.M., and W.-K.W.; supervision, K.-Y.W., M.M., and W.-K.W.; project administration, K.-Y.W. and W.-K.W. All authors have read and agreed to the published version of the manuscript.

**Funding:** This research received no external funding.

**Acknowledgments:** For financial support, the third author wishes to thank the Australian Research Council and the National Science Council, Ministry of Science and Technology (MOST), Taiwan. The fourth author acknowledges the Research Grants Council of Hong Kong (Project Number 12500915), Ministry of Science and Technology, Taiwan (MOST, Project Numbers 106-2410-H-468-002 and 107-2410-H-468-002-MY3), Asia University, China Medical University Hospital, and Hang Seng Management College. The fourth author would also like to thank Robert B. Miller and Howard E. Thompson for their continuous guidance and encouragement.

**Conflicts of Interest:** The authors declare no conflict of interest.

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
