# Peer review of "Review on Efficiency and Anomalies in Stock Markets"

_economies, doi:10.3390/economies8010020_

Round 1

Reviewer 1 Report

This paper is the review paper that examines the theory and literature on market efficiency and anomalies. However, I confused after careful reading of this paper. What is the motivation for this review paper? How many readers take time to read this paper carefully? Authors should be more concise and clear.

If authors want to publish this paper, major revise is absolutely required.

My suggestions are

Using table, which can summarize the key issue and visualize the category in section 2 just before 2.6 (Explaining Market Efficiency in Subdividing Areas). Authors should more focus on recent empirical results in 2.6 section. This paper considers very rich in references, but it seems that too many references. You would better to consider major papers. The section 4 of “Behavioral Finance” seems to less relevant to this topic. I strongly recommend to delete this portion.

Author Response

Thank you very much for your invaluable comments and suggestions, which have improved the revised version significantly.

  • This paper considers very rich in references

Many thanks.

Below are our responses to your helpful comments and suggestions.

 Question 1:  What is the motivation for this review paper? How many readers take time to read this paper carefully? Authors should be more concise and clear.

Answer 1:  We have emphasized the motivation for the review paper in the revised version. We have addressed the issue as to how many readers might read the paper carefully, and explained in the abstract and Introduction of the revised paper why academics, investors and policy makers might wish to read it.

Question 2: Using table, which can summarize the key issue and visualize the category in section 2 just before 2.6 (Explaining Market Efficiency in Subdividing Areas).

 Answer 2:  We have used tables to summarize the key issue and visualize the category in Section 2, and explained Market Efficiency in Subdividing Areas in the revised paper. We have added a table to summarize the Fama (1970) taxonomy of the EMH tests, and added a table to summarize different factor models in the revised paper.

 Question 3: Authors should more focus on recent empirical results in 2.6 section.

 Answer 3:  We have focused on the recent empirical results in Section 2.6 in the revised paper.

 Question 4: This paper considers very rich in references, but it seems that too many references. You would better to consider major papers.

Answer 4:  We have emphasized the major papers in the revised paper, and deleted some less important papers.

Question 5: The section 4 of "Behavioral Finance" seems to less relevant to this topic. I strongly recommend to delete this

 Answer 5:  "Behavioral Finance" is very important to the topic of the paper. The last comment of Reviewer 2 supports this view, so we would prefer to retain this section.

Reviewer 2 Report

Title

Review on Efficiency and Anomalies in Stock Markets

Manuscript ID

economies-688801

Summary: This paper approaches from a theoretical point of view the financial literature on market efficiency and the various anomalies found as well as the theories developed around them. To do so, authors firstly revise the concept of market efficiency and delve into the Efficient-Market Hypothesis (EMH), considering opposing visions that have arisen over time. Subsequently, they comment on several market anomalies and review different theories within the Behavioral Finance field that can be used to offer an explanation for such deviations from efficiency.

Authors suggest that this literature review might be of interest for (1) academics studying EMH, market anomalies and other topics related to Behavioral Finance; (2) investors interested in making informed decisions on their investment; and (3) policy makers reviewing market-related policies.

Overall comments: The object of study is appropriate – Economic Behavior and Behavioral Finance has proved to be relevant in economics research, as reflected by several recent Nobel prices in economics as 2002, 2013, 2017 and 2019, among others.

In this sense, implications of heuristics and cognitive biases in economic decision-making can be better approached from a behavioral perspective, at the junction where economics meets psychology. Thus, reviewing stock market anomalies from such perspective is doubtlessly significant. However,

1) authors give excessive attention to quantitative models instead of trying to contextualize the whys and wherefores around such behaviors – the latter would be of most interest and help to consolidate powerful economic-related behavioral insights.

2) Although it is stated by the authors that their intention is to provide useful information for investors and policy makers, it seems not clear how this objective is articulated throughout the paper. I would suggest adopting a more precise guiding thread to avoid readers losing track throughout multiple subsections and ensure a clear and concise message is sent after the literature review.

3) Accordingly, and given the length of the article, I would recommend reformulating section four to unambiguously state conclusions and, in so doing, emphasize the added value offered by the authors with this research work. It would similarly be advisable to use summary tables when a lot of citations have to be written (see [1267-1277], [1447-1454] and [1477-1481] as examples of too long citations that distort reading).

In general terms, the research paper has potential to be published although (i) authors should ensure a guiding thread is clearly noticeable by the reader; (ii) non-relevant sections have to be shortened, especially those devoted to provide a detailed explanation of quantitative issues; (iii) summary tables must be used to display information and enhance readers’ understanding; and (4) behavioral finance and its roots must deeply impregnate the perspective by which the subject of study is approached, for example, better connecting opposing visions around efficiency and behavioral finance.

Author Response

Thank you very much for your invaluable comments and suggestions, which have improved the revised version significantly.

  • The object of study is appropriate - Economic Behavior and Behavioral Finance has proved to be relevant in economics research, as reflected by several recent Nobel prices in economics as 2002, 2013, 2017 and 2019, among others.
  • In this sense, implications of heuristics and cognitive biases in economic decision-making can be better approached from a behavioral perspective, at the junction where economics meets psychology. Thus, reviewing stock market anomalies from such perspective is doubtlessly significant.
  • In general terms, the research paper has potential to be published

Many thanks.

Below are our responses to your helpful comments and suggestions.

 Question 1:  authors give excessive attention to quantitative models instead of trying to contextualize the whys and wherefores around such behaviors - the latter would be of most interest and help to consolidate powerful economic-related behavioral insights.

 Answer 1:  We have reduced the excessive attention to quantitative models in the revised paper.

Question 2: Although it is stated by the authors that their intention is to provide useful information for investors and policy makers, it seems not clear how this objective is articulated throughout the paper. I would suggest adopting a more precise guiding thread to avoid readers losing track throughout multiple subsections and ensure a clear and concise message is sent after the literature review.

 Answer 2:  We have emphasized how the review provides useful information for investors and policy makers in the revised paper. We have adopted a more precise guiding thread to focus the revised version more clearly. We have addressed the issue as to how many readers might read the paper carefully, and explained in the abstract and Introduction of the revised paper why academics, investors and policy makers might wish to read it.

Question 3: Accordingly, and given the length of the article, I would recommend reformulating section four to unambiguously state conclusions and, in so doing, emphasize the added value offered by the authors with this research work.

 Answer 3:  We have reformulated Section 4 to state the conclusions clearly, and have emphasized the value added in the revised version.

Question 4: It would similarly be advisable to use summary tables when a lot of citations have to be written (see [1267-1277], [1447-1454] and [1477-1481] as examples of too long citations that distort reading).

Answer 4:  We have used tables to summarize the key issue and visualize the category in Section 2, and explained Market Efficiency in Subdividing Areas in the revised paper. We have added a table to summarize the Fama (1970) taxonomy of the EMH tests, and added a table to summarize different factor models in the revised paper.

Question 5: authors should ensure a guiding thread is clearly noticeable by the reader;

Answer 5:  We have emphasized a guiding thread in the revised paper.

Question 6: non-relevant sections have to be shortened, especially those devoted to provide a detailed explanation of quantitative issues;

Answer 6:  We have shortened the non-relevant sections in the revised paper.

Question 7: summary tables must be used to display information and enhance readers' understanding;

Answer 7:  We have used summary tables to display information more clearly in the revised paper.

Question 8: behavioral finance and its roots must deeply impregnate the perspective by which the subject of study is approached, for example, better connecting opposing visions around efficiency and behavioral finance.

 Answer 8:  "Behavioral Finance" is very important to the topic of the paper. The last comment of Reviewer 2 supports this view, so we would prefer to retain this section.

Round 2

Reviewer 1 Report

Happy with the response by the author and the changes made.